# Mitochondria-derived peptide SHLP2 regulates energy homeostasis through the activation of hypothalamic neurons

Seul Ki Kim[1,2], Le Trung Tran[1,2], Cherl NamKoong[3], Hyung Jin Choi [3,4], Hye Jin Chun[5], Yong-ho Lee [5], MyungHyun Cheon[6], ChiHye Chung [6], Junmo Hwang[7], Hyun-Ho Lim[7], Dong Min Shin[1,2], Yun-Hee Choi[1] & Ki Woo Kim [1,2] ✉

Small humanin-like peptide 2 (SHLP2) is a mitochondrial-derived peptide implicated in several biological processes such as aging and oxidative stress. However, its functional role in the regulation of energy homeostasis remains unclear, and its corresponding receptor is not identified. Hereby, we demonstrate that both systemic and intracerebroventricular (ICV) administrations of SHLP2 protected the male mice from high-fat diet (HFD)-induced obesity and improved insulin sensitivity. In addition, the activation of pro-opiomelanocortin (POMC) neurons by SHLP2 in the arcuate nucleus of the hypothalamus (ARC) is involved in the suppression of food intake and the promotion of thermogenesis. Through high-throughput structural complementation screening, we discovered that SHLP2 binds to and activates chemokine receptor 7 (CXCR7). Taken together, our study not only reveals the therapeutic potential of SHLP2 in metabolic disorders but also provides important mechanistic insights into how it exerts its effects on energy homeostasis.

Mitochondria are unique organelles of mammalian cells that possess their own distinct circular mitochondrial DNA (mtDNA). This DNA is known to contain 37 genes that encode 2 rRNAs, 22 tRNAs, and 13 polypeptide subunits of the oxidative phosphorylation (OXPHOS) enzyme complexes[1]. Recent studies have identified that mtDNA contains additional short open reading frames (sORFs)2 that encode several small regulatory peptides[2]. These peptides are collectively named mitochondrial-derived peptides (MDPs)[3]. Increasing evidence indicates that MDPs are widely distributed in various tissues throughout the body and play crucial roles in numerous biological activities[3–5].

One of the first MDPs discovered is Humanin, a 24-amino-acid peptide encoded by the mitochondrial 16S ribosomal RNA (rRNA)

gene. It was originally identified in 2001 during a screening for neuroprotective factors against Alzheimer's disease[6,7]. A number of subsequent studies over the past two decades have extensively demonstrated the cytoprotective and beneficial metabolic effects of humanin[3,5,8]. Another MDP called Mito chondrial Open Reading Frame of the 12S rRNA-c (MOTS-c) is encoded by the mitochondrial 12S rRNA gene[9]. MOTS-c is a mitochondria-originated metabolic hormone which regulates cellular metabolism and promotes energy homeostasis[9]. MOTS-c is also known to communicate with the nuclear genome to modulate nuclear gene expressions under metabolic stress[10] and it acts as an exercise-induced factor to enhance physical capacity[11]. Moreover, it has been shown that humanin and MOTS-c promote lifespan, while

[1]Division of Physiology, Department of Oral Biology, Yonsei University College of Dentistry, Seoul 03722, Korea. [2]Department of Applied Life Science, BK21 FOUR, Yonsei University College of Dentistry, Seoul 03722, Korea. [3]Neuroscience Research Institute, Seoul National University College of Medicine, Seoul 03080, Korea. [4]Department of Biomedical Sciences, Seoul National University College of Medicine, Seoul 03080, Korea. [5]Department of Internal Medicine, Yonsei University College of Medicine, Seoul 03722, Korea. [6]Department of Biological Sciences, Konkuk University, Seoul 05029, Korea. [7]Neurovascular Unit Research Group, Korea Brain Research Institute (KBRI), Daegu 41068, Korea. ✉e-mail: kiwoo-kim@yuhs.ac

their plasma levels tend to decline with age[8,11]. This suggests that the MDPs are also potential therapeutic targets for numerous age-related metabolic disorders.

In addition to humanin and MOTS-c, a recent study has identified an additional six small humanin-like peptides (SHLP1 to 6) encoded by the 16S rRNA gene of the mtDNA genome[4]. Among these, SHLP2 has been reported to have cytoprotective and metabolic effects in mouse models, suggesting that there may exist a potential receptor for SHLP2 in mice[4,12,13]. Moreover, while both SHLP2 and SHLP3 have been shown to enhance cellular differentiation and insulin sensitivity in adipocytes in vitro, only SHLP2 was found to improve insulin action in vivo[4]. However, the mechanism by which SHLP2 affects whole-body energy homeostasis has not been investigated. Furthermore, the cellular receptor for SHLP2 and its primary site of action have not been identified.

In the present study, we discovered that serum SHLP2 levels were significantly lower in obese and diabetic patients compared to healthy individuals. In murine animal models, administration of SHLP2 resulted in the suppression of food intake, activation of thermogenesis, and prevention of diet-induced obesity. Additionally, a combination of experimental approaches, including patch-clamp techniques and the designer receptors exclusively activated by designer drugs (DREDD) system, revealed that POMC neurons in the hypothalamic ARC are necessary for the SHLP2-induced suppression of food intake and increased energy expenditure (EE). Using high-throughput structural complementation screening, we identified a potential receptor for SHLP2.

## Results

### Serum SHLP2 levels are decreased in obese and diabetic patients
SHLP2 is associated with insulin sensitivity in mice[4]. This prompted us to investigate whether the serum SHLP2 levels might be altered by metabolic disease conditions in human. To answer this question, we first developed a polyclonal antibody for SHLP2 and validated its specificity (Supplementary Fig. 1a–c). In addition, our antibody was able to specifically detect SHLP2 without any cross-reactivity with other SHLPs, highlighting its high specificity for SHLP2 (Supplementary Fig. 1d). Subsequently, we conducted dot blot analyses using human sera collected from normal, obese, and diabetic subjects (Supplementary Table. 1). Interestingly, we observed a significant decrease in serum levels of SHLP2 in both diabetic and obese patients compared to healthy individuals (Fig. 1a). The decreased levels of serum SHLP2 were also observed in murine obesity and diabetic models, such as *ob/ob* and *db/db* mice (Supplementary Fig. 1e), indicating that the serum levels of SHLP2 may be influenced by individual metabolic conditions.

### SHLP2 protects mice from diet-induced metabolic syndrome
To investigate the metabolic effects of SHLP2, we administered various doses of SHLP2 intraperitoneally (IP) and identified the effective dose (2 mg/kg, IP) for acute suppression of food intake (Supplementary Fig. 2a). At this dose, SHLP2 exhibited the unique ability to suppress rebound food intake and body weight gain for up to 8 h after refeeding, as compared to scrambled peptides, highlighting the specific role of SHLP2 in regulating energy homeostasis (Supplementary Fig. 2b, c). Nonetheless, under normal chow (NC) conditions, SHLP2 only exhibited a tendency to decrease food intake and body weight (Supplementary Fig. 2d–f). Notably, when SHLP2 was injected daily for 3 weeks, it improved glucose tolerance and insulin sensitivity, suggesting its intrinsic effects on insulin sensitivity independent of body weight (Supplementary Fig. 2g, h).

In contrast to NC, SHLP2 significantly protected mice against diet-induced body weight gain in high-fat diet (HFD) conditions (Fig. 1b). In addition, SHLP2 injection decreased total body fat mass and circulating leptin levels (Fig. 1c, d). Histological analysis further revealed a

significant reduction in the size of inguinal and epididymal white adipose tissues (WAT), as well as a profound decrease in HFD-induced hepatic steatosis, as evidenced by decreased intracellular lipid droplets and reduced expression of lipogenic genes in hepatocytes (Fig. 1e–h). Moreover, the administration of SHLP2 resulted in a significant lowering of blood glucose levels and improved glucose tolerance and insulin sensitivity (Fig. 1i–k). Taken together, these results demonstrated that the systemic SHLP2 administration protects mice against HFD-induced obesity and metabolic disorders.

### SHLP2 suppresses food intake and promotes thermogenesis
To understand the underlying mechanisms of how SHLP2 influences metabolism, we utilized indirect calorimetry to assess metabolic parameters following a four-week period of systemic SHLP2 administration. Systemic SHLP2 administration robustly increased $O_2$ consumption ($VO_2$), $CO_2$ production ($VCO_2$) as well as heat generation without alteration in the physical activities (Fig. 2a–d). Given that the interscapular brown adipose tissue (iBAT) is a well-known organ for thermogenesis, primarily through the action of uncoupling protein 1 (UCP1)[14], we further examined the thermogenic effect of SHLP2 in the iBAT. SHLP2 markedly up-regulated *Ucp1*, as well as the genes involved in iBAT thermogenesis including *Pgc1α*, *Dio2*, *Prdm16*, and *Nrf1* (Fig. 2e). Moreover, the iBAT of SHLP2-administered mice displayed a decrease in lipid vacuoles, an increase in UCP1 protein levels, and a higher mitochondrial content compared to mice injected with saline (Fig. 2f, g). The increase in iBAT thermogenesis in the SHLP2-injected mice may be attributed to an increase in sympathetic tone directed to the iBATs. This was proven by the elevation of plasma norepinephrine (NE) level and the activation of the cAMP response element-binding protein (CREB) and p38 MAPK, which are the downstream effectors of the β3-adrenergic receptor signaling in the iBATs (Fig. 2h, i)[15]. In addition to a notable increase in EE, mice treated with SHLP2 showed a significant reduction in daily food intake (Fig. 2j). This decrease in food intake was linked to a reduction in the expression of orexigenic neuropeptides, including agouti-related peptide (*Agrp*) and neuropeptide Y (*Npy*), in the hypothalamus (Fig. 2k). These results indicate that systemic administration of SHLP2 prevented the diet-induced obesity through an enhanced thermogenesis and a decrease in energy uptake.

### SHLP2 activates the hypothalamic neurons
The diverse effects of SHLP2 on iBAT thermogenesis, elevated serum NE levels, as well as its anorexigenic impact by modulating hypothalamic neuropeptides, suggested that the CNS could potentially serve as a target site for SHLP2 and play a crucial role in its metabolic effects. To investigate whether SHLP2 can directly reach the CNS, we systemically injected SHLP2 or saline and probed for SHLP2 in the cerebrospinal fluid (CSF). Remarkably, SHLP2 was highly detected in the CSF of SHLP2-administered mice (Fig. 3a). Next, we evaluated c-Fos immunoreactivity in various brain regions following systemic injection of SHLP2. The results showed that the mediobasal hypothalamus, a major region in the brain that regulates metabolic homeostasis, exhibited the highest c-Fos activation (Fig. 3b, c)[16,17]. Analysis of c-Fos immunoreactivity throughout the mediobasal hypothalamus revealed a significant increase in the arcuate nucleus (ARC) and dorsomedial hypothalamus (DMH) in the SHLP2-injected mice, while no differences were observed in the ventromedial nucleus (VMH) and lateral hypothalamus (LH) between the saline and SHLP2 groups (Fig. 3b–d). In contrast to the hypothalamus, we did not observe any significant changes in c-Fos activation in other brain regions such as the cortex, hippocampus, and periaqueductal gray (PAG) following systemic administration of SHLP2 (Supplementary Fig. 3a–c). The ICV administration of SHLP2 into the third ventricle (3 V) showed similar results with distinct c-Fos activation mainly in the ARC and DMH (Fig. 3e, f). These findings suggested that SHLP2 could cross the brain-CSF barrier and directly activate hypothalamic neurons.

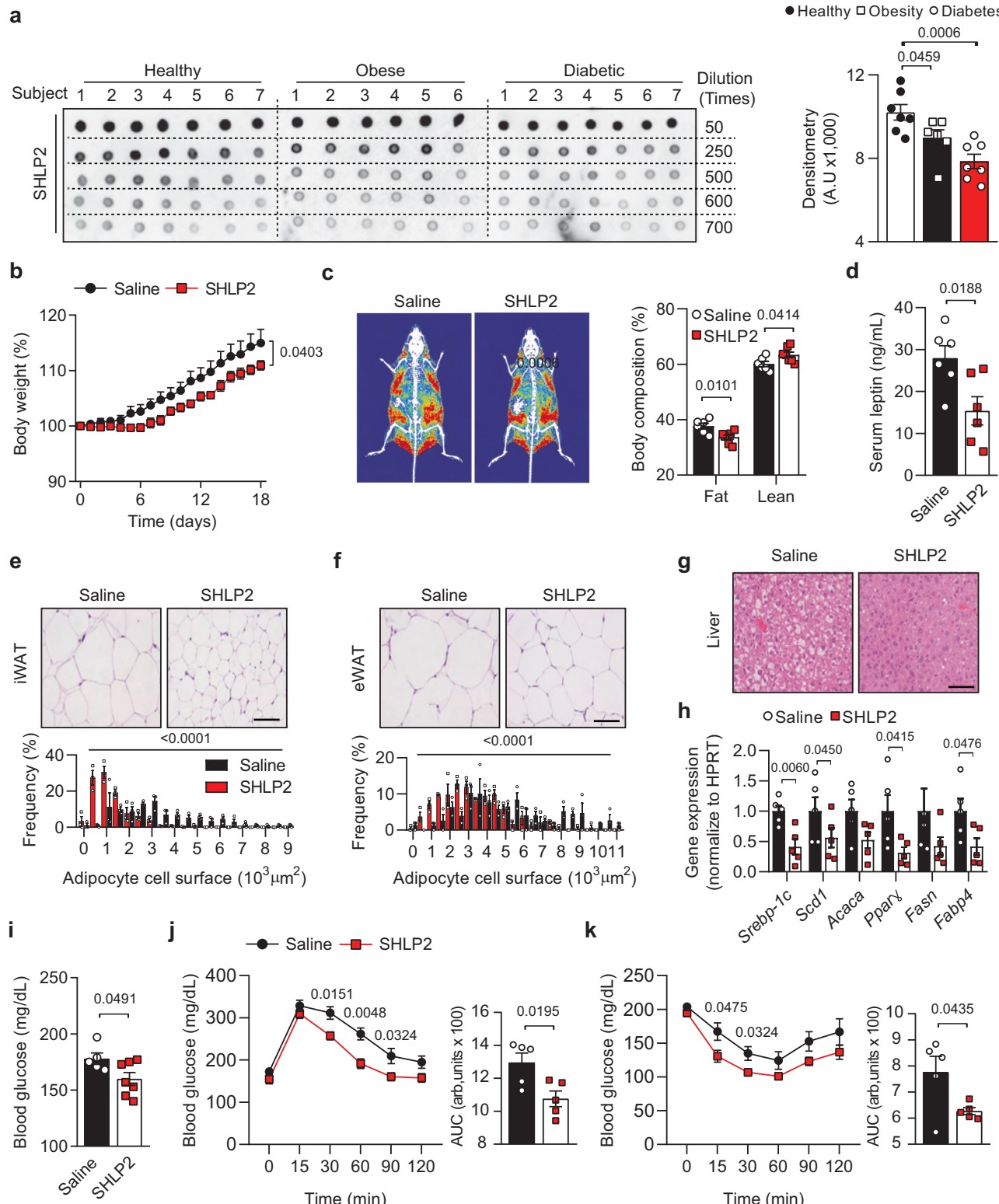

**Fig. 1 | Protective effect of SHLP2 from HFD-induced obesity. a** Dot-blot image (left) and relative blot densitometry (right) detecting plasma SHLP2 levels in healthy ($n = 7$), obese ($n = 6$), and diabetic ($n = 7$) male subjects. **b** Change of body weight (%) in male mice fed with HFD after daily intraperitoneal (IP) injection of saline or SHLP2 for 3 weeks ($n = 9$ saline, $n = 11$ SHLP2). **c** Representative DEXA images (left) and body composition (right) ($n = 6$ saline, $n = 7$ SHLP2). **d** Serum leptin levels ($n = 6$ each group). **e, f** Representative adipose tissues images from iWAT (**e**) and eWAT (**f**) ($n = 3$). Frequency distribution of adipocytes from iWAT and eWAT samples (bottom graphs). **g** Representative liver images ($n = 3$). **h** Expression of lipogenic genes in liver ($n = 5$ each group). **i** Blood glucose levels in fed condition ($n = 5$ saline, $n = 7$ SHLP2). **j, k** GTT (**j**) and ITT (**k**) were performed in HFD-fed male mice before body weight divergence following IP administration of either saline or SHLP2 ($n = 5$ each group). Scale bars, 50 µm. $n$ indicates the number of biologically independent human subjects or animals examined. Data were presented as mean ± SEM. Two-tailed Student's $t$ tests were used in (**a**), (**c**–**k**), and two-way ANOVA with Bonferroni post-hoc tests were used in (**b**). Source data are provided as a Source Data file.

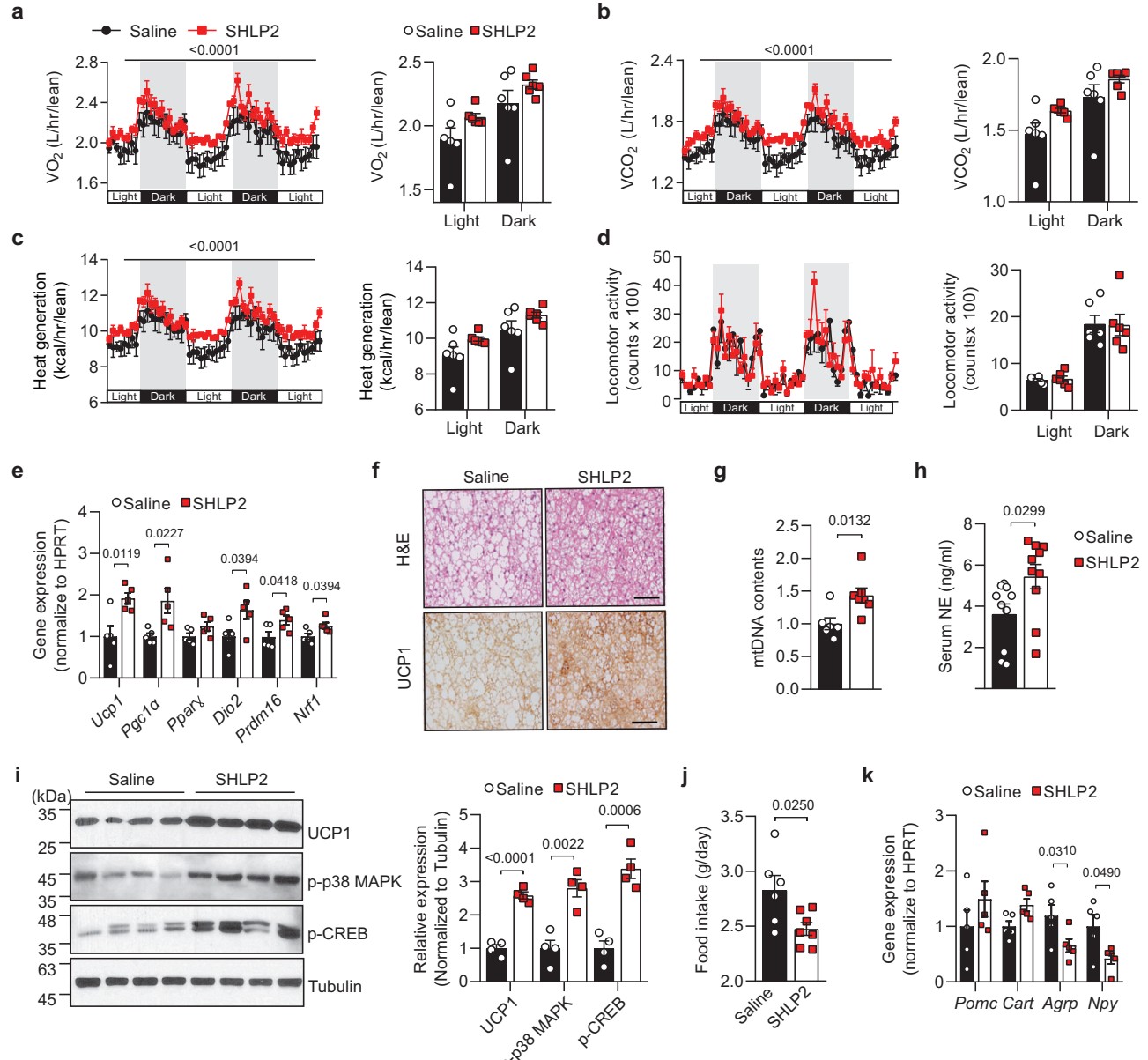

**Fig. 2 | SHLP2 enhances iBAT thermogenesis and suppresses energy intake.**
**a**–**d** Temporal changes (left) and average values (right) of $O_2$ consumption (**a**), $CO_2$ production (**b**), heat generation (**c**), and locomotor activity (**d**) from HFD-fed male mice with IP administration of either saline or SHLP2 ($n = 6$ each group). **e** Expression of thermogenic genes in iBAT from indicated groups ($n = 5$ each group). **f** Representative H&E (top) and UCP1 (bottom) staining from iBAT of HFD-fed male mice ($n = 3$ each group). **g** Mitochondrial DNA contents in the iBAT ($n = 6$ saline, $n = 7$ SHLP2). **h** Serum norepinephrine (NE) levels in indicated group ($n = 10$ each group). **i** Immunoblots (left) and relative densitometry (right) of UCP1, p-p38 MAPK, p-CREB expression in the iBAT samples ($n = 4$ each group). Data shown are representative of three independent experiments with similar results. **j** Daily food intake of HFD-fed male mice treated with either saline or SHLP2 ($n = 6$ saline, $n = 7$ SHLP2). **k** Gene expression of neuropeptides regulating feeding in the hypothalamic samples ($n = 5$ each group). Scale bars, 50 μm. $n$ indicates the number of biologically independent animals examined. Data were presented as mean ± SEM. Two-tailed Student's $t$ tests were used in bar graphs, and two-way ANOVA with Bonferroni post-hoc tests were used in line graphs. Source data are provided as a Source Data file.

## Central SHLP2 administration improved energy homeostasis

Based on the aforementioned results, we hypothesized that SHLP2 directly affects hypothalamic neuronal circuits and whole-body energy homeostasis. To examine this possibility, we administered SHLP2 directly into the 3 V and assessed metabolic changes (Supplementary Fig. 4a). Following analysis of dose responses, we determined that a 3ug was appropriate for ICV injection (Supplementary Fig. 4b, c). Interestingly, a single ICV injection of SHLP2 produced a suppressive impact on food intake and body weight gain for up to three days, in both NC and HFD-fed states (Fig. 4a, b, and Supplementary Fig. 4d–f). Following ICV administration of SHLP2, a consistent increase in EE was observed without any change in locomotor activity, even in NC-fed

conditions (Fig. 4c–f). The expression of thermogenic genes and UCP1 was significantly enhanced in iBAT (Fig. 4g, h). Additionally, SHLP2 improved glucose tolerance and exhibited a strong trend towards increasing insulin sensitivity (Fig. 4i, j and Supplementary Fig. 4g, h). These results strongly indicate that the thermogenic and anorexigenic effects of SHLP2 might be mediated through the CNS.

## Activation of POMC neurons is involved in the anti-obesity effect of SHLP2

To investigate which neuronal populations are responsible for the anorexigenic and thermogenic effects of SHLP2, we monitored POMC and AgRP neurons in the ARC, which regulate food intake and EE[18-20].

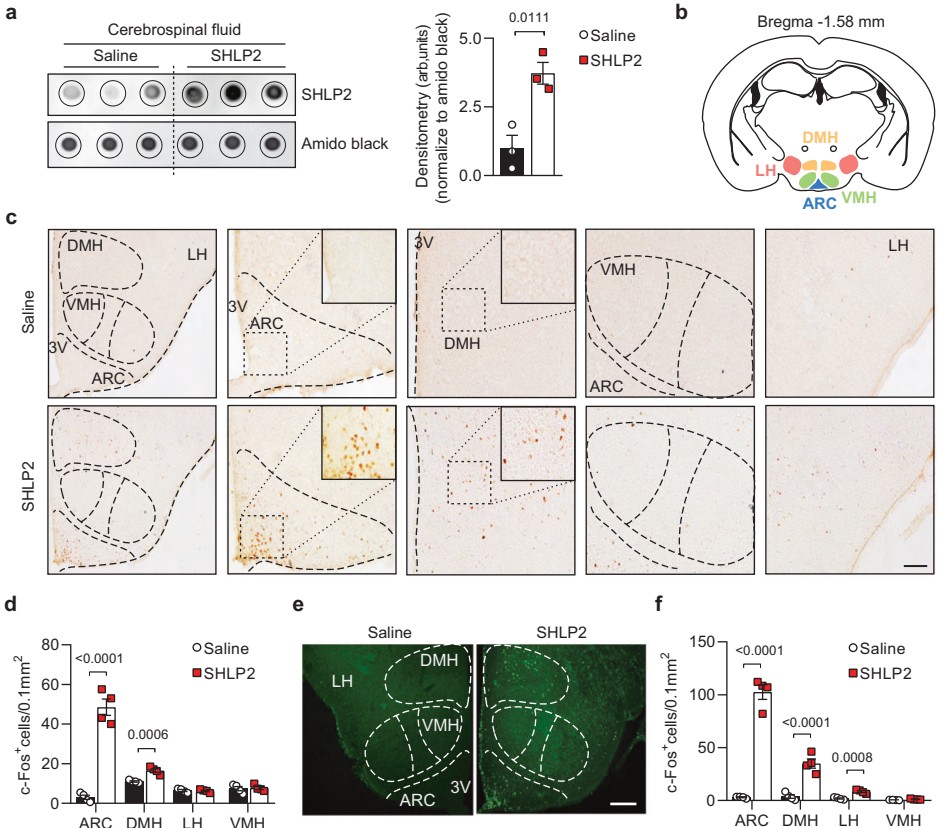

**Fig. 3 | SHLP2 induces c-Fos activation in hypothalamic neurons. a** Dot blots (left) for SHLP2 and their relative densitometry (right) in CSF after an IP injection of saline or SHLP2 ($n = 3$ each group). **b**, **c** c-Fos immunoactivity in mediobasal hypothalamic nuclei after an IP injection of saline or SHLP2. **d** Quantification of c-Fos activities from (**c**) ($n = 4$ each group). **e**, **f** Representative figures (**e**) and graph quantification (**f**) of c-Fos immunoreactivity in hypothalamic regions after an ICV administration of saline or SHLP2 ($n = 4$ each group). Scale bars, 100 μm. $n$ indicates the number of biologically independent animals examined. Data were presented as mean ± SEM. Two-tailed Student's $t$ tests were used in bar graphs. Source data are provided as a Source Data file.

Following SHLP2 administration, we monitored c-Fos activation in POMC or AgRP neurons using POMC-Cre or AgRP-ires-Cre mice with the Cre-dependent reporter tdTomato, allowing for the visualization of POMC and AgRP neurons, respectively (tdTomato^POMC-Cre or tdTomato^AgRP-Cre mice)[20–23]. Upon ICV injection of SHLP2, c-Fos expression was primarily observed in POMC neurons (51.4%) as opposed to AgRP neurons (22.8%) (Fig. 5a and Supplementary Fig. 5a). Subsequently, whole-cell patch clamp experiments were conducted on POMC neurons to further examine the effects of SHLP2 on their activity (Fig. 5b–e). In response to SHLP2 (1 μM), 34.49% of POMC cells (10 out of 29) were depolarized, resulting in a mean change of +4.43 ± 2.31 mV in the resting membrane potential (Fig. 5b–e). The depolarized POMC cells were distributed across the mediobasal hypothalamus and were clustered near the 3 V (Supplementary Fig. 5b). Interestingly, the depolarization of POMC neurons induced by SHLP2 was not affected by the presence of synaptic blockers such as TTX, picrotoxin, CNQX, and D-AP-V in the bath solution, as evidenced by an increase in resting membrane potential by +3.17 ± mV in 4 out of 8 tested cells (Fig. 5f, g). However, the SHLP2 application resulted in only a minor suppression of AgRP neuron activity (Supplementary Fig. 5c, d). These results strongly suggest that SHLP2 directly activates POMC neurons while marginally inhibiting AgRP/NPY neurons.

To further confirm the significance of POMC neuronal activation in the metabolic effects mediated by SHLP2, we employed two different DREDD systems. First, we performed stereotaxic injection of adeno-associated virus (AAV) that expressed either a Cre-dependent mCherry (pAAV-mCherry) or an mCherry reporter fused with the hM4Di receptor (pAAV-hM4D(Gi)-mCherry) into the hypothalamus of POMC-Cre mice. This allowed us to generate POMC-mCherry or POMC-hM4Di mice. Second, we produced mice that selectively expressed the hM4Di receptor solely in POMC neurons (hM4Di^POMC-cre) by crossing POMC-Cre mice with mice that carried the floxed hM4Di allele[21,24]. By utilizing these two distinct methods, we were able to selectively suppress the activity of POMC neurons through treatment with Clozapine-N-oxide (CNO) (Fig. 5h, i and Supplementary Fig. 5e–i). Control mice with intact POMC neurons, such as POMC-mCherry and hM4Di^F/F, demonstrated a significant decrease in rebound food intake and body weight gain upon ICV administration of SHLP2 (Fig. 5j, k and Supplementary Fig. 5j, k). However, when POMC neuron activity was blocked by CNO treatment in the POMC-hM4Di and hM4Di^POMC-cre mice, the suppressive effects of energy intake by SHLP2 were notably blunted (Fig. 5j, k and Supplementary Fig. 5j, k). The thermogenic effects of SHLP2 were evident in control mice, as demonstrated by the marked increase in skin and rectal temperature. Nonetheless, inhibiting POMC neurons compromised these SHLP2-induced thermogenic effects (Fig. 5l–n). The results support the involvement of hypothalamic POMC neurons in the regulation of food intake and energy expenditure mediated by SHLP2.

## The chemokine receptor CXCR7 mediates effects of SHLP2

To identify the cellular receptor(s) responsible for SHLP2 activity, we conducted a screening using the cell-based PathHunter β-Arrestin assay [https://www.discoverx.com]. This assay is capable of monitoring GPCR activation in a homogeneous and non-imaging manner, utilizing enzyme fragmentation complementation with β-galactosidase as the functional reporter. A total of 168 known GPCRs were screened[25].

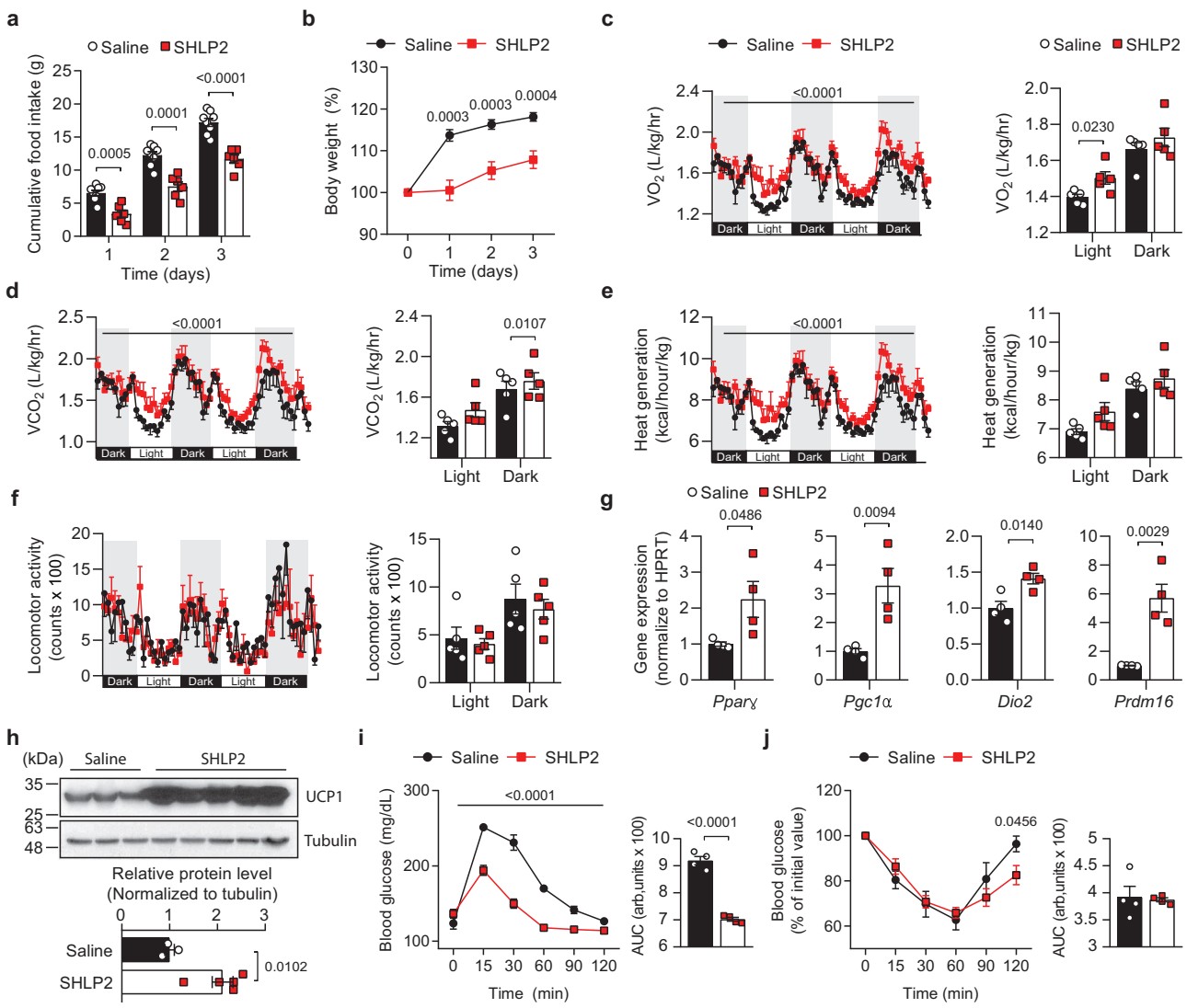

**Fig. 4 | Central SHLP2 regulates energy homeostasis. a, b** Cumulative food intake (**a**) and percent body weight change (**b**) after an ICV administration of saline or SHLP2 ($n = 8$ saline, $n = 6$ SHLP2). **c–f** Temporal changes (left) and average value (right) of $O_2$ consumption (**c**), $CO_2$ production (**d**), heat generation (**e**), and locomotor activity (**f**) of male mice after an ICV injection of saline or SHLP2 ($n = 5$ each group). **g** Expression of thermogenic genes in the iBAT ($n = 4$ each group). **h** Immunoblots (top) for UCP1 and relative densitometry (bottom) in the iBAT ($n = 3$ saline, $n = 5$ SHLP2). Data shown are representative of three independent experiments with similar results. **i, j** GTT (**i**) and ITT (**j**) were performed in NC-fed male mice after 4 hrs of ICV administration of saline or SHLP2 ($n = 4$ each group). $n$ indicates the number of biologically independent animals examined. Data were presented as mean ± SEM. Numbers of animals examined were expressed in parenthesis. Two-tailed Student's $t$ tests were used and two-way ANOVA with Bonferroni post-hoc tests were used in (**c–f**), (**i**). Source data are provided as a Source Data file.

(Fig. 6a). Out of the screened GPCR candidates, SHLP2 showed the highest efficiency in recruiting β-Arrestin to the chemokine receptor CXCR7 (previously known as ACKR3). The relative agonistic activity of SHLP2 was approximately 70% of its known ligand, stromal cell-derived factor-1 (SDF1α)/CXCL12 (Fig. 6b and Source Data Fig. 6b). By applying the same experimental paradigm, we found that SHLP2 induced β-Arrestin recruitment to CXCR7 in a dose-dependent manner, with a half maximal effective concentration ($EC_{50}$) of 0.9696 μM (Fig. 6c). Additionally, the SHLP2 binds directly to CXCR7, as evidenced by the detection of tagged CXCR7 through pulldown analysis (Fig. 6d). The SHLP2 has a specific effect on the CXCR7 chemokine receptor, causing its internalization, without affecting CXCR4 (Supplementary Fig. 6a–g). This suggests that SHLP2 has a selective affinity for CXCR7, and it could potentially activate distinct signaling pathway through this receptor. The interaction between SHLP2 and CXCR7 was further confirmed using the Nanoluc-complementation system, which

demonstrated the recruitment of β-Arrestin to CXCR7[26] (Fig. 6e). SHLP2-induced recruitment of β-Arrestin to CXCR7 was comparable to the known ligand CXCL12 at a concentration 50 ng/mL (Fig. 6f).

CXCR7 is known to activate the MAP Kinase - ERK1/2 signaling cascade by recruiting β-Arrestins upon ligand binding[27–29]. Thus, the SHLP2 could potentially induce ERK1/2 phosphorylation and subsequently increase the excitation of POMC neurons[30,31]. Indeed, the SHLP2 treatment resulted in strong phosphorylation of ERK1/2 in vitro (Fig. 6g). To confirm the necessity of ERK signaling for the depolarizing effect of SHLP2 on POMC neurons, acute hypothalamic slices were pre-incubated with the MAP kinase blocker PD98059 after confirming CXCR7 expression in the neurons (Supplementary Fig. 6h). The blockage of MAPK/ERK signaling abolished the SHLP2-induced depolarization of all recorded POMC neurons (Fig. 6h, i, k). On the other hand, blocking PI3K, which is a converging point for many peptides and hormones on POMC neurons[32], did not affect the depolarizing

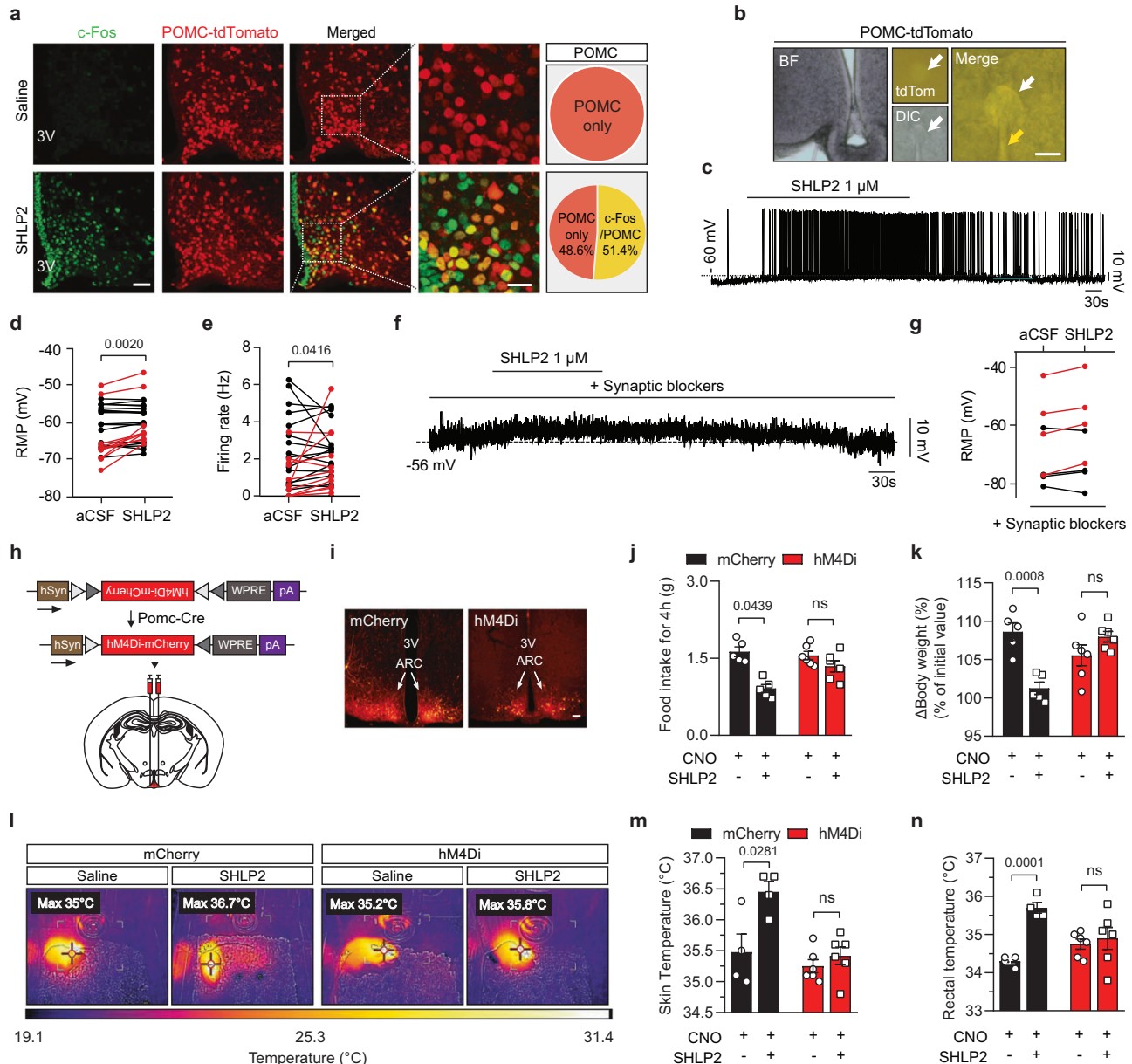

**Fig. 5 | SHLP2 activates POMC neurons and it requires for metabolic effects of SHLP2. a** Percentage of c-Fos activation in POMC neurons after an ICV administration of SHLP2 (*n* = 3 each group). Scale bars, 20 μm. **b** POMC neurons are targeted using fluorescent (tdTomato) or differential interference contrast (DIC) microscopy in the tdTomato^POMC-Cre male mice. White and yellow arrows indicate the targeted cell and patch pipette. Scale bars, 10 μm. **c–e** Representative depolarization response (**c**) of POMC neurons and effects on resting membrane potential (RMP) (**d**) and firing rate (**e**) of a subset of POMC neurons (*n* = 27 neurons from 11 animals, red: cells showing an increase in RMP more than 2 mV). **f, g** Representative depolarizing response (**f**) and summary graph (**g**) of POMC neurons under the pretreatment of synaptic blockers (red: cells showing an increase in RMP more than 2 mV). **h** Design of AAV virus (top) bilaterally injected into the ARC. **i** mCherry fluorescence confirming the viral targets of POMC neurons in the ARC of POMC-cre

male mice. Scale bars, 100 μm. hSyn, Human synapsin 1 promoter; hM4Di-mcherry, Gi-coupled hM4D DREADD fused with mCherry; WPRE, woodchuck hepatitis posttranscriptional regulatory element; pA, poly(A) tails. **j, k** Cumulative food intake (**j**) and ΔBody weight (%) (**k**) after ICV administration of SHLP2 in the POMC-mCherry or POMC-hM4Di male mice (*n* = 5 each mCherry group, *n* = 6 each hM4Di group). **l, m** Representative thermal images (**l**) and average skin temperature (**m**) measured by thermoscanner. **n** Effect of SHLP2 on rectal temperature (*n* = 4 each mCherry group, *n* = 6 each hM4Di group). *n* indicates the number of biologically independent animals examined. Data were presented as mean ± SEM. Two-tailed Student's *t* tests were used in bar graphs. Wilcoxon matched-pairs signed rank test were used in electrophysiological result. Source data are provided as a Source Data file.

effect of SHLP2 (Fig. 6j, k). Therefore, our data suggested that CXCR7 is a receptor for SHLP2 and mediates its effects through the MAPK-ERK1/2 signaling cascade.

## Discussion

Our current study provides insights into the specific mechanisms and regulatory roles of SHLP2 in the maintenance of energy homeostasis.

In addition, the identification of a potential receptor for SHLP2's cellular activities in mice sheds further light on the underlying molecular pathways involved.

Importantly, both systemic and ICV administration of SHLP2 in mice provided protection against diet-induced obesity, resulting in reductions in body weight and fat mass, as well as improvements in insulin sensitivity. Intriguingly, we found that SHLP2 could cross the

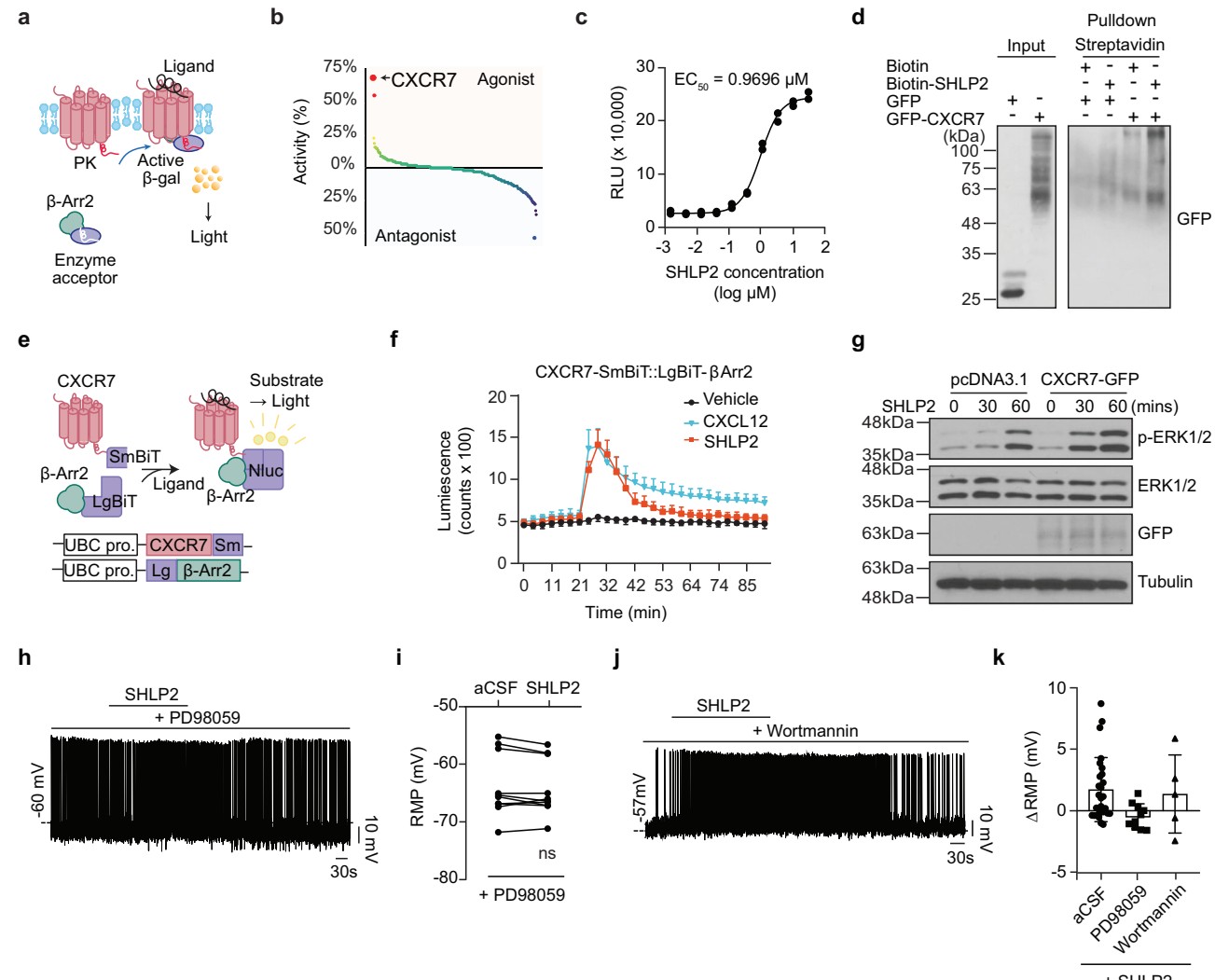

**Fig. 6 | Chemokine receptor CXCR7 interacts with SHLP2. a** Schematic for PathHunter β-Arrestin assay. **b** Activity plot showing the results of PathHunter β-Arrestin GPCR screening. Note that CXCR7 exhibited the highest activity by SHLP2. **c** Dose-dependent β-Arrestin recruitment to CXCR7 by SHLP2. The concentration of SHLP2 inducing half-maximal response ($EC_{50}$) was calculated by least-square fitting to a four-parameter logistic curve. **d** Pulldown analysis of streptavidin-SHLP2-biotin showing the interaction between SHLP2 and CXCR7. Data shown are representative of five independent experiments with similar results. **e** Schematic (upper panel) and the plasmid DNA (lower panel) of the NanoBiT complementation assay to monitor SHLP2-induced β-Arrestin2 recruitment to CXCR7. **f** Results of β-Arrestin2 recruitment to CXCR7 by SHLP2 or CXCL12 ($n = 4$). **g** Western blot result of time-dependent ERK1/2 phosphorylation by SHLP2 with or without CXCR7 overexpression. Data shown are representative of five independent experiments

with similar results. **h** Representative spike of POMC neuron showing the depolarization by SHLP2 is blocked by the pretreatment of the MAP Kinase blocker PD98059. **i** Summary graph of SHLP2 effect on the resting membrane potential (RMP) of POMC neurons under treatment of the MAP Kinase blocker ($n = 10$ from 4 animals; ns: not significant, paired $t$ test). **j** Representative depolarizing response of POMC neurons by SHLP2 under treatment of the PI3K inhibitor wortmannin. **k** Summary graph showing the changes in resting membrane potential (ΔRMP) by SHLP2 treatment with or without MAP Kinase or PI3K inhibitor ($n = 27, 10, 5$, respectively). $n$ indicates the number of biologically independent cells examined. Data are presented as the mean ± SD. PK, enzyme donor fragment ProLink; β-Arr2, β-Arrestin2; β-gal, β-galactosidase; Nluc Nano Luciferase, SmBiT Small fragment, LgBiT Large fragment, UBC pro UBC promotor. Source data are provided as a Source Data file.

brain-CSF barrier and activate the POMC neurons of the ARC, which directly suppressing food intake and promoting thermogenesis. Moreover, our study identified CXCR7 as a cellular receptor for SHLP2, providing detailed insight into the mechanism underlying its pharmacological actions. These findings suggest that SHLP2 may represent a promising therapeutic target for metabolic disorders, and offer potential avenues for the development of effective treatments for these conditions.

Metabolic disorders and other diseases can cause a decline in mitochondrial function, leading to a gradual loss of MDP expressions and potentially reducing their functions[33–35]. Our findings suggest that this may also be the case for SHLP2, as we observed decreased serum levels of SHLP2 in both diabetic and obese patients, indicating that it

may have a preventative role in human metabolic diseases (Fig. 1a). Animal models of diabetes or obesity, such as *db/db* and *ob/ob* mice, also exhibited decreased levels of SHLP2, further supporting this hypothesis. Several previous reports supported the notion that the MDPs might play important roles in preventing several diseases. For example, humanin was decreased in human diseases such as Alzheimer's disease, mitochondrial encephalopathy, lactic acidosis, and stroke-like episodes[8]. In addition, the circulating MOTS-c was reduced in diabetes patients[36]. Interestingly, circulating SHLP2 was decreased in aged mice, implying that SHLP2 might be involved in the aging process of rodents[4]. Therefore, it would be interesting to explore whether certain MDPs, including SHLP2, can be utilized as diagnostic markers for specific human diseases such as metabolic or age-related disorders.

Following intraperitoneal administration of SHLP2 in mice, there was a reduction in appetite and food intake and an increase in brown fat thermogenesis (iBAT). Moreover, SHLP2 was observed in the cerebrospinal fluid (CSF) and activated c-Fos in multiple hypothalamic neurons, including the arcuate nucleus (ARC), dorsomedial hypothalamus (DMH), and lateral hypothalamus (LH), indicating that the brain could be a potential target for SHLP2's metabolic regulation. Immunohistochemistry and electrophysiological approaches, indeed, identified POMC neurons in the ARC as primary targets for the action of SHLP2. The ex vivo patch-clamp experiments revealed that SHLP2 depolarized a subset of hypothalamic POMC neurons, and the activation did not rely on synaptic inputs. Specifically, the fraction of neurons was activated under the treatment of TTX and synaptic blockers, indicating that SHLP2 acted directly on the POMC neurons, independent of synaptic inputs. Further experiments using inhibitory DREADDs targeting POMC neurons confirmed that SHLP2 regulates the hypothalamic melanocortin system. The heterogeneity of POMC neurons in the arcuate nucleus, characterized by different expressions of ion channels and receptors, may result in diverse neuronal responses to hormones and neurotransmitters[37–39]. Thus, future research identifying ion channels or related receptors responsible for SHLP2-mediated excitability of POMC neurons holds promise for future investigation. Additionally, since activation of POMC neurons has been linked to the regulation of blood pressure and cardiac function[40], it would be interesting to investigate whether SHLP2 is also involved in the regulation of other intrinsic functions of POMC neurons, such as blood pressure and cardiac function. In contrast to its effect on POMC neurons, SHLP2 slightly inhibited AgRP neurons. Additionally, our observation on SHLP2-induced c-Fos expression in other hypothalamic neuronal populations suggests that SHLP2 may also play a role in regulating other physiological functions in the central nervous system beyond POMC neurons.

The pharmacological action of SHLP2 was further revealed by the identification of CXCR7 as a receptor mediating its action. We used a GPCR screening system to identify potential receptors. The chemokine receptor CXCR7 appeared as a potential receptor for SHLP2. Further characterizations showed that SHLP2 recruits β-Arrestin 2 at the concentration of inducing POMC-neuron activation. CXCR7 and β-Arrestin 2 are known to activate the MAP Kinase–ERK cascade[28,41], thus we hypothesized that SHLP2 induced the depolarization of POMC neurons through ERK1/2 phosphorylation. Indeed, the MAP kinase inhibition blunted the effect of SHLP2 on the tested POMC neurons. Although the identification of CXCR7 as the receptor of SHLP2 in activating ERK signaling and POMC neuronal activation is a significant finding, further investigation is necessary to fully understand the exact mechanism of SHLP2. Specifically, additional studies are needed to explore whether SHLP2 affects other signaling pathways in addition to ERK. By investigating broader pharmacological effects of SHLP2, we may gain a more comprehensive understanding of its potential as a therapeutic target for various diseases. Interestingly, previous studies have demonstrated that CXCR7 is involved in the modulation of ER stress in cardiac cells through AMPK activation[42]. This raises a possibility that genetic approaches targeting the CXCR7-encoding *Ackr3* gene (CXCR7 KO) in vivo might be a way to elucidate a more complete signaling cascade of CXCR7 and SHLP2. By using genetic approaches, we may be able to better understand the interplay between CXCR7 and other pathways and ultimately develop more effective therapeutic strategies targeting this signaling pathway.

In summary, our current study has demonstrated that the mitochondrial-derived peptide SHLP2 has profound metabolic benefits through its actions on the central nervous system, specifically on POMC neurons in the hypothalamus. In addition, identifying the activation of the chemokine receptor CXCR7 by SHLP2 highlights its potential as a novel therapeutic candidate for the treatment of metabolic diseases.

## Methods

### Reagents
Small humanin-like peptide 1 to 6 (SHLP1-6) were purchased from AnyGen [http://www.anygen.com/eng/]. For each peptide, the Fmoc-Ser (tBu)-Wang resin was allowed to swell in dimethylformamide (DMF). The amino acid was then deprotected using a solution of 20% piperidine in DMF. Coupling was carried out using Hexafluorophosphate Benzotriazole Tetramethyl Uronium (HBTU), *N*-methylmorpholine (NMM) in DMF. Stepwise deprotection and coupling of amino acids was repeated until the desired peptides was synthesized. The peptides were cleaved from dried resin using a trifluoroacetic acid (TFA) solution containing 2.5% 1, 2 ethanedithiol (EDT), 5% thioanisole and 5% distilled water. Crude peptides were precipitated in ether, and dried under vacumm. Peptides were purified using commercial columns (YMC-Triart C18/S-5 μm/12 nm.5μm (20 x 250 mm) and each fraction was obtained at a flow rate of 1 mL/min. The collected fractions were analyzed via HPLC (Shimadzu HPLC LabSolution) and MALDI-TOF MS (AXIMA Assurance, Shimadzu). The fractions containing the pure peptide were mixed and then lyophilized. SHLP2 were freshly prepared at 1 μg/μL by diluting in saline, and were freshly used for mice. The information of SHLP peptide sequence used in study were listed in Supplementary Table 3.

### Antibody generation
Custom rabbit anti-SHLP2 were generated from Ab Frontier [http://www.younginfrontier.com/laboratory/abfrontier/]. Two (one for each SHLP2 peptide synthesized from Ab Frontier) healthy New Zealand white rabbits (1.8–2.0 kg, obtained from the Sinyang) were subcutaneously immunized with 1 mL of the prepared SHLP2 peptide. The rabbits were then boosted twice by the same SHLP2 peptide on day 14, 28 and were sacrificed on day 35 for all anti-serum collection. To sacrifice the mice, general anesthesia was induced by intraperitoneal injection of Zoletil (0.5 mg/10 g), followed by heart puncture for exsanguination for blood collection.

### Human subject
Ethical approval was obtained from the Committee of the Yonsei University College of Medicine (4-2017-1168)[43] for the collection of human peripheral blood samples and performed in accordance with the Helsinki declaration. All participants provided written informed consent. The information of human subject used in study were listed in Supplementary Table. 1.

### Animals
All animal experiments and surgical procedure were approved by the Institutional Animal Care and Use Committee (IACUC) of the Avison Biomedical Research Center, Yonsei University. Male mice of 8-12 weeks of age were used in all experiments and housed in individual cages with a 12:12 h light–dark cycle (light automatically on/off at 08:00-20:00) at 22 ± 1 °C and 55% humidity. Mice were fed with either regular chow diet (Lab Diet, 5053) or high fat diet (Research Diets, D12492) and provided with water *ad libitum*. For the experiments, C57BL/6J (JAX No. 000664)[44], C57BL/6J-*ob/ob* (SLC, Inc. Tokyo, Japan), C57BLKS/J-*db/db* (JAX No. 000662)[45], POMC-Cre (JAX No. 005965)[21], AgRP-Ires-Cre (JAX No. 012899)[20], R26-tdTomato (JAX No. 007914)[22], R26-hM4Di/ mCitrine (JAX No. 026219)[24] mice were used. POMC-Cre mice were crossed with R26-tdTomato or R26-hM4Di/mCitrine mice to generate POMC-Cre/ R26-tdTomato (tdTomato^POMC-Cre) or POMC-Cre/ R26-hM4Di/mCitrine (hM4Di^POMC-Cre) mice. AgRP-Ires-Cre mice were crossed with R26-tdTomato mice to generate AgRP-Ires-Cre/ R26-tdTomato (tdTomato^AgRP-Cre).

### Surgery
For intracerebroventricular (ICV) cannulation, mice were anesthetized by intraperitoneal injection (IP) of Zoletil (titelamine/zolazepam

30 mg/kg) and Rompun (xylazine 10 mg/kg), then placed onto a stereotaxic apparatus. After setting the ventricular coordinates, a permanent 26-gauge stainless steel cannula (Plastic one, Inc.) was implanted in the third ventricle (1.8 mm caudal to bregma, and 5.0 mm ventral to the sagittal sinus according to the Paxinos and Franklin's atlas) and secured to the skull surface using a dental acrylic (Vertex). Mice were individually housed and allowed to recover for at least 1 week. The angiotensin II-induced water consumption test was performed to confirm the position of the ICV cannulation. An immediate dipsogenic reaction within the first 10 min was observed in the animals with the correct cannula placement. Mice with no angiotensin II-induced thirst behavior were excluded from the experiments.

For pharmacogenetic studies to target the POMC neurons, pAAV-hSyn-DIO-hM4D(Gi)-mCherry (Addgene, 44362) or pAAV-hSyn-DIO-mCherry (AAV2) control (Addgene, 50459) were bilaterally injected into the hypothalamic ARC of 10-week-old POMC-cre mice. Mice were anesthetized using Zoletil/Rumpun and were placed into a stereotaxic apparatus (David Kopf Instruments, Kopf 1900). The skull was exposed via a small incision, and a small burr hole was drilled. A Nanofil 10 μL syringe with a 35 G blunt-end needle was inserted into the brain for virus delivery. Adenovirus injected via a syringe pump (KD Scientific) at a rate of 50 nL/min for 5 min. 0.25 μL per injection site of ARC using appropriate coordinates: anteroposterior (AP), −1.8 mm; medio-lateral (ML), ±0.25 mm; dorso-ventral (DV), −5.7 mm. Two weeks following the virus injection, a cannula was implanted into the third ventricle and the angiotensin II-induced water consumption test was performed to confirm the correct cannula placement. Mice with incorrect stereotaxic injection outside the ARC were excluded from analysis after confirmation of mCherry expression.

## SHLP2 injection

For IP injections, SHLP2 and control saline were administered daily at 2 mg/kg. After 3 weeks of injections, the body compositions were analyzed using Dual-energy X-ray absorptiometry (DEXA) system (InAlyzer™, MEDIKORS) and metabolic assessments were performed.

For ICV injection, mice were handled daily for 3 days to minimize stress responses and administered with 3 μg of SHLP2 or control saline followed with overnight fasting (18 h) and metabolic assessments were evaluated after 24 h of injection.

For chemogenetic inactivation of POMC neurons, mice were ICV administration of 1 μg of CNO (Tocris, 4936). The metabolic study with SHLP2 or saline was performed 1 h after the CNO injection. Acute food intake was measured after ICV injection of Saline of SHLP2 followed with overnight fasting.

## Metabolic phenotyping

Metabolic cage study was performed using the 12-channel Pheno-Master Home Cage System (TSE Systems) equipped with triple beam IR technology to assess indirect calorimetry, food and water consumption, oxygen consumption ($VO_2$) carbon dioxide production ($VCO_2$), and local motor activity[46,47]. After acclimation for 48 h, physiological parameters were monitored for a 96-h recording period. Average metabolic parameters from three 24-h acquisition cycles were calculated and statistically analyzed. The relationship between energy expenditure and total body mass was normalized to either lean body mass or body weight$^{0.75}$. We used the latter because lean mass could not be measured due to the ICV cannula. Body fat and lean masses of mice were analyzed by a DEXA system (InAlyzer™, MEDIKORS)[46]. The DEXA system was calibrated in accordance with the operator's instructions before measurement and was set as the lower limit detection of 0.001 g/cm$^2$.

## Rectal and skin temperature

Body temperatures were monitored after SHLP2 or saline ICV injections with 1-h pre-treatment of CNO at room temperature (RT). Rectal and skin temperatures of mice were measured using a Thermalert TH-5 thermometer (Physitemp Instruments) with a small-diameter temperature probe or an FLIR-E5XT handheld infrared camera (FLIR Systems). Several infrared pictures were taken of the mice during the light phase while it was allowed to move freely in the house cage. Skin temperature was recorded by storing the temperature of the warmest pixel in thermal images.

## Glucose and insulin tolerance tests

Glucose tolerance (GTT) and insulin tolerance (ITT) tests were performed in mice after 3 weeks of IP injection or 4 h of ICV injection of SHLP2. For GTT, mice were fasted overnight with water *ad libitum*. The ITT were performed under fed condition. The D-glucose (1 g/kg of body weight, Sigma-Aldrich, G8270) or insulin (1.2 U/kg of body weight, Humulin R, Eli Lilly) were administered by IP injection. Blood glucose levels were measured from a tail-nick venous blood using a handheld glucose monitor (Contour TS, Ascensia Diabetes Care). Blood glucose levels were measured before (t = 0) and after glucose or insulin injections at indicated time points.

## Acute brain slice preparation

The 6- to 12-week-old tdTomato$^{POMC-Cre}$ or tdTomato$^{AgRP-Cre}$ or hM4Di$^{POMC-Cre}$ mice were anesthetized with IP injections of Zoletil/Rumpun. Mice were rapidly perfused with ice-cold cutting artificial cerebrospinal fluid (aCSF) comprising of (in mM) 93.0 N-methyl-D-glucamine, 2.5 KCl, 1.2 $NaH_2PO_4$, 24.0 $NaHCO_3$, 20.0 HEPES, 7.5 Glucose, 5.0 Na-ascorbate, 3.0 Na-pyruvate, 10 $MgSO_4$ and 0.5 $CaCl_2$, with pH of 7.3-7.4 and an osmolality of 300-330 mOsm/kg. The brain was dissected and a block containing the hypothalamus was made. Two to three 250-300 μm thick coronal slices containing the ARC of the hypothalamus were cut on 7000smz-2 Vibratome (Campden Instrument) in cold cutting aCSF. Slices were moved to carbogenated recording aCSF containing (in mM) 124.0 NaCl, 2.5 KCl, 1.2 $NaH_2PO_4$, 24.0 $NaHCO_3$, 5.0 HEPES, 5.0 Glucose, 2.0 $MgSO_4$ and 2.0 $CaCl_2$, pH 7.3−7.4, and osmolality 300-310 mOsm/kg. Slices were left to recover for 45 min at 35 °C, then at least 1 h at RT before any further experiments.

Slices were moved to a submerged-type recording chamber with continuous perfusion of carbogenated recording aCSF heated to 31 °C at a flow rate of approximately 2 mL/min. Tested compounds were added through bath perfusion. To block synaptic inputs, the aCSF contained 100 nM tetrodotoxin (TTX citrate; Tocris, 1069), 100 μM picrotoxin (Sigma-Aldrich, P8390), 100 μM D(−)-2-Amino-5-phosphonopentanoic acid (D-AP-V; Tocris, 0106) and 20 μM CNQX (CNQX-disodium; Tocris, 1045). To block ERK1/2 activation or PI3K signaling pathway, 10 μM of PD98059 (Tocris, 1213) or 1 μM of wortmannin (Tocris, 1232) was added to the bath solution for at least 30 min before any measurement. To test the effect of CNO on POMC neurons expressing the DREDD hM4Di receptor, 5 μM of CNO were applied for 1 to 2 min.

## Whole-cell patch-clamp recording

POMC or AgRP-expressing neurons in the ARC were identified with a brief exposure to epifluorescence on an upright microscope (Nikon FN1) using a 40X water-immersion objective and differential interference contrast optics. The pipette solution contains (in mM) 127 K-Methanesulfonate, 5 KCl, 10 HEPES, 0.3 EGTA, 4 Mg-ATP, 0.3 Na-GTP, and is of a pH adjusted to 7.3 with KOH and a 288 mOsm/kg osmolality. The patch pipettes have a resistance of 4-6 MΩ when filled with the pipette solution. Whole-cell configuration was achieved with brief negative pressure at a holding voltage of -50 mV. The basal resting membrane potential and firing rate were recorded for around 3 min after breaking in, and bath perfusion of SHLP2 was applied for 3 to 5 min. Neurons were classified as responsive if there was a change in resting membrane potential of at least 2 mV during the period of drug

application[48]. The access resistance ($R_a$) was monitored throughout the experiment, and cells with a $R_a$ of more than 35 MΩ were excluded from the analysis. Data were acquired at 10 kHz, filtered with a lowpass filter at 5 kHz and analyzed offline with Clampfit 10 (Molecular Devices). The calculated liquid junction potential was +9.55 mV and used to correct all data.

## Hormone measurements

Serum insulin and leptin levels were measured by ELISA kits (Morinaga Institute of Biological Science, M1104 and M1305, respectively) in accordance with the manufacturer's instructions.

For serum norepinephrine (NE), commercial ELISA kits (Labor Diagnostika Nord GmbH & Co., BA-E-5200) were used following the manufacturer's manual. Serum was extracted by using a cis-diol-specific affinity gel and acylated, then converted enzymatically before being quantitatively determined by a competitive enzyme immunoassay method.

## Immunohistochemistry (IHC)

Whole brains were dissected out and post-fixed with 10% neutral buffered formalin overnight and dehydrated in 20% sucrose solution until the tissue sank to the bottom of the container. The brains were frozen-sectioned into 25 μm slices in LEICA section system (Leica Biosystems, SM2010R) and stored in anti-freeze solution before used. Brain sections were washed and then incubated 30 min in 1X PBS containing 0.25% (v/v) Triton X-100 (PBT). Brain sections were blocked in 3% goat serum prepared in 0.1% PBT for 1 h at room temperature (RT) and then incubated overnight with c-Fos antibody (Cell Signaling, 2250, 1:1000) or DsRed antibody (TaKaRa, 632496, 1:200) or CXCR7 antibody (Proteintech, 20423-1-AP, 1:500) at 4 °C. Washed brain sections were incubated in the goat anti-rabbit secondary antibody with Alexa Fluor 488 (Invitrogen, A21206, 1:1000) or Alexa Fluor 594 (Invitrogen, A11012, 1:1000) prepared in PBT containing 3% goat serum. Sections were washed and mounted on glass slides. Brain slides were visualized using a LEICA fluorescence microscope (LEICA Corporation) or a confocal laser microscope (LSM700) with Cal Zeiss Zen 2.6 imaging software. c-Fos were counted using Fiji (image J2) 2.3.0 and Adobe Photochop CS6 software.

For UCP1 staining, paraffin-embedded sections of formalin-fixed iBAT samples were mounted on a glass slide and incubated with UCP1 primary antibody (Abcam, ab10983, 1:1000) overnight at 4 °C in a humidified chamber. The slides were washed 5 times with 1X PBS and then incubated with biotinylated donkey anti-rabbit IgG (Jackson ImmnuoResearch, 711-065-152, 1:1000) for 2 h at RT, followed by an hour incubation in a solution of avidin-biotin complex (Vector, PK-6200). After 2 washes, the slides were stained with 3,3'-diaminobenzidine (DAB)-peroxidase substrate solution (0.04% DAB and 0.01% $H_2O_2$) in PBS for less than 2 min and were visualized by a Nikon Digital Camera DXM1200 microscope system (Nikon Corporation).

## Cell culture

HEK293T cells (ATCC) were cultured in Dulbecco's modified Eagle medium (DMEM; Capricorn scientific) supplemented with 10% fetal bovine serum (FBS; Corning), 100 U/mL penicillin, and 100 μg/mL streptomycin (Gibco). Cells were maintained at 37 °C and in a humidified, 5% $CO_2$ atmosphere.

## Immuno-dot blot assay

The immuno-dot blot assay was performed by using the human and mice serum or mouse CSF. 0.2 μm nitrocellulose membranes were activated with ultrapure water (Invitrogen, 10977015), and 2 μL of samples were spotted on a membrane and dried for 30 mins at RT. The membrane was blocked in 5% BSA prepared in 1X Tris Buffered Saline-0.1% Tween 20 (TBST) for 1 h before being overnight incubated with anti-SHLP2 at 4 °C. Membranes were washed 3 times with 1X TBST,

then incubated with anti-rabbit IgG (Invitrogen, 31460, 1:3000). The membranes were washed and visualized using the ImageQuant™ LAS 4000 (GE Healthcare Life Science). Blot densitometry was measured using ImageJ software (NIH Image). To quantitate the immuno-dot blot assay results, we prepared controls (total protein level stained by amido black), and subsequently established a calibration figure for each run of the assay.

## Pull-down assay

Ten μg of the plasmid DNA pEGFP-N1 (Clonetech) or pcDNA3.1-Zeo-hCXCR7-sGFP (a gift of Prof. Jong-Ik Hwang, Korea University)[49] were transfected to HEK293T cells using Lipofectamine 2000 (Invitrogen, 11668019) according to the manufacturer's instruction. After the transfections for 24 h, cells were trypsinized and collected in ice-cold 1X PBS. The cell suspension was incubated with either 10 μM Biotinylated SHLP2 or Biotin (Sigma-Aldrich, B4501) for 30 min at 4 °C, then cross-linked by 1 mM of disuccinimidyl suberate (DSS; Thermo Scientific) for 30 min. Cells were washed with Tris-buffered saline (pH 7.5) to quench and remove excess cross-linkers and lysed in RIPA buffer. To capture the biotinylated peptide-receptor complex, streptavidin agarose beads (Sigma-Aldrich, 21655) were added and incubated at 4 °C for 4 h. Beads were collected by centrifugation and washed with RIPA buffer. Elution was performed with 1% SDS and 100 mM Dithiothreitol (DTT) for 1 h at RT. Fractions were subjected to routine polyacrylamide gel electrophoresis and immunoblotting.

## Western blot

Tissue or cell lysates were prepared in RIPA buffer as previously describe[46]. Equal protein amounts were to SDS-PAGE electrophoresis. Separated proteins on gels were then transferred onto PVDF membranes, blocked in 5% skim milk for 1 h before being incubated overnight at 4 °C. In the following primary antibodies: UCP1 (Abcam, ab10983, 1:10,000), p-p38 MAPK (Cell Signaling, 4511, 1:1000), p-ERK (Cell Signaling, 9101, 1:1000), ERK (Cell Signaling, 9102, 1:1000), p-CREB (Cell Signaling, 9196, 1:1000), GFP (Sigma-Aldrich, G1544, 1:3000) and α-tubulin (Developmental Studies Hybridoma Bank, 12G10, 1:5000). Membranes were washed with 1X TBST and then were incubated with anti-rabbit IgG (Invitrogen, 31460, 1:3000) or anti-mouse IgG (Invitrogen, 62-6520, 1:2000). Blots were visualized by ImageQuant™ LAS 4000 or X-ray film. Blots densitometry were measured using ImageJ software.

## Quantitative PCR (q-PCR)

Total RNA was extracted from homogenized tissues using the TRizol reagent (Thermo Scientific) and equal amounts of total RNA were reversely transcribed into cDNA using ReverTra Ace qPCR RT Master Mix with gDNA Remover (Toyobo, FSQ-301). The cDNA templates were then amplified and quantified by the BioRad CFX Real-Time PCR system (BioRad) using iQ™ SYBR® Green Supermix (BioRad, 1708880).

To measure mitochondrial DNA content, total DNA from iBAT samples was extracted using AccuPrep® Genomic DNA Extraction Kit (Bioneer, K-3032) according to the manufacturer's instructions. Total DNA from each sample was amplified using primers specific for the mitochondrial cytochrome C oxidase subunit 2 (*Cox2*) gene and the ribosomal protein s18 (*Rps18*) nuclear gene. The primer sequences used in qPCR were listed in the Supplementary Table 2.

## Internalization assay

HEK293T cells were seeded onto the glass coverslips coated with Poly-L-Lysine and transfected with 1 μg of pcDNA3.1-Zeo-CXCR7-SGFP or pcDNA31-Zeo-CXCR4-SGFP. After 24 h, cells are treated with 10 μM SHLP2 or DMSO at indicated time points. Cells were fixed with cold methanol and incubated with anti-GFP antibody overnight at 4 °C. The GFP were visualized with anti-rabbit-AlexaFluor-488 secondary antibody, and the nuclei were stained with DAPI. Cells were examined

using laser-scanning microscope LSM700. At least 100 cells were counted from each replication.

## GPCR screening
The GPCR profiling was performed using PathHunter GPCRmax Arrestin recruitment assay with 168 known GPCRs provided by DiscoverX (Eurofins). This assay utilized Chinese Hamster Ovary CHO-K1 cells expressing GPCR tagged with ProLink (PK) and β-Arrestin2 tagged with a deletion mutant of β-galactosidase as an enzyme acceptor. Upon the binding of the ligand to the GPCR, the recruitment of tagged β-Arrestin2 to the GPCR causes the complementation of a functional β-galactosidase, hydrolyzing a chemiluminescence substrate. The assay was performed with known ligands in agonist and antagonist modes, with data presented as:

$$\%\text{Activity (agonist)}$$
$$= 100\% \times \frac{\text{Mean RLU (SHLP2)} - \text{Mean RLU (vehicle control)}}{\text{Mean MAX RLU (control ligand)} - \text{Mean RLU (vehicle control)}}$$

and

$$\%\text{Activity (antagonist)}$$
$$= 100\% \times \left(1 - \frac{\text{Mean RLU (SHLP2)} - \text{Mean RLU (vehicle control)}}{\text{Mean MAX RLU (control ligand)} - \text{Mean RLU (vehicle control)}}\right)$$

The dose response ($EC_{50}$) was built using a 10-point curve and fit to a Hill Equation.

## Nanoluc-complementation based β-Arrestin2 recruitment assay
The recruitment assay was performed based on Nano-luciferase[49]. The vectors (50 ng; gift of Prof. Jong-Ik Hwang, Korea University) encoding human CXCR7, CXCR4, and β-Arrestin2 tagged with SmBiT and LgBiT under Ubiquitin C promoter[49] were transfected into HEK293T cells in a 96-well clear bottom culture plate using Lipofectamine 2000 according to the manufacturer's protocol. Cells were washed three times with an assay buffer containing (in mM) 140 NaCl, 2.5 KCl, 10 Glucose, 5 HEPES, 2 CaCl$_2$, and 2 MgSO$_4$, pH 7.4 after 24 h. Cells were then incubated with 5 μM Coelenterazine-h (Promega, S201) in the assay buffer for 30 min in the dark, and luminescence was measured at 35 °C in the Victor Nivo Multimode Plate reader (PerkinElmer). The background luminescence was measured for at least 10 min before the addition of ligands, and signals were monitored for at least 60 min after the application of SHLP2 or recombinant mouse CXCL12 (R&D systems, 460-SD-010/CF). The specificity of SHLP2 β-Arrestin2 recruitment through CXCR7 was also confirmed through the assay with CXCR4. Luminescent signals were collected through a 700 nm-IR filter and data were shown as light counts per each cycle of measurement.

## Statistics
Prism 8.0 (GraphPad Software) was used for all the statistical analyses and making graphs. Data are represented as mean ± SEM and a p-value less than 0.05 was defined as a statistically significant difference. Unpaired 2-tailed Student's t tests or one-way analysis of variance (ANOVA) with Tukey's post-tests were used to determine the statistical significance between two or multiple groups, respectively. Two-way ANOVA with the Bonferroni post-tests was used to determine the statistical significance between multiple groups with two independent variables. Detailed statistical test(s) used and corresponding p-value are described in figure legends. Differences were considered statistically significant at *$P < 0.05$, ** $P < 0.01$; ***$P < 0.001$.

## Reporting summary
Further information on research design is available in the Nature Portfolio Reporting Summary linked to this article.

## Data availability
The datasets generated in this study are available in the Supplementary Information/Source Data file. Supplementary Information/Source data is included with this paper.

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

## Acknowledgements

This research was supported by the Korea Drug Development Fund, funded by the Ministry of Science and ICT, the Ministry of Trade, Industry, and Energy, and the Ministry of Health and Welfare (HN22C0645, Republic of Korea). The research was also supported by the National Research Foundation of Korea (2021R1A2C4002011 and RS-2023-00217709 for K.W.K; 2021R1I1A1A01049540 for Y.-H.C) and the Korea Health Industry Development Institute (Korea Health Technology R&D Project: HR22C141102 for K.W.K). This work was also partly supported by the KBRI basic research program (23-BR-01-02 for H.-H.L.).

## Author contributions

K.S.K. performed experiments, analyzed data, wrote, and finalized the manuscript. T.L.T. performed whole-cell patch-clamp recording and edited the manuscript. C.NK., H.J.C., M.H.C., C.H.C., J.M.H., and H.H.L. assisted with experimental design and analysis. H.J.J. and Y.H.L. provide human samples. Y.H.C. and D.M.S. reviewed and edited the manuscript. K.W.K. conceptualized the research, edited, and finalized the manuscript.

## Competing interests

The authors declare no competing interests.
