## [Peer Review File · Nature Communications]

Mitochondria-derived peptide SHLP2 regulates energy homeostasis through the activation of hypothalamic neuronsREVIEWER COMMENTS

Reviewer #1 (Remarks to the Author):

This study investigated the role of SHLP2, a mitochondrial gene derived peptide, in DIO and the potential underlying mechanism. The data presented showed that SHLP2 reduced DIO, improved glucose metabolism, and activated a subset of brain neurons, including POMC neurons. Further studies suggested that it may be mediated by CLCX7 receptors. Although SHLP2 has been previously suggested to produce beneficial effects on metabolism, this study further extended that by examining its effect on DIO and with brain-specific effects. The experiments appeared to be vigorously investigated with all necessary controls and data were also appropriately analyzed. However, the evidence with POMC neurons as the mediating mechanism seems not be strong.

1) The study proposed that SHLP2 inhibited feeding through activation of feeding. However, previous studies including the one Zhan C et al, J. Neurosci. 2013 suggest that optogenetic activation of POMC neurons fails to reduce feeding, especially within the a few hours time window. Other studies with Gq Dredd also showed the same result. Thus, it is perplexing that SHLP2 activation of POMC neurons was able to produce such an impressive effect on reducing feeding within a 4 hours period. The authors need to conduct additional experiments to clarify this critical issue.

2) It is also important that the effects of SHLP2 may be mediated by other neurons that are activated by SHLP2, as the authors showed additional brain regions within the Arc as well as other brain sites with c-Fos activation by SHLP2. Importantly, some of the effect of SHLP2 may be mediated by neuron inhibition, which is suggested by, for example, reduced AgRP/NPY shown in this manuscript, but can't be detected with c-Fos used here. Thus, the effect on POMC neurons seems to be, at most, one component of the mechanism.

3) Although the biology of SHLP2 on CLCX7 was well characterized, how it is related to POMC neurons is completely lacking. Thus, the description on mechanism related to POMC neurons is not well supported by the data.

4) The effect on glucose metabolism was shown in DIO mice with existing body weight difference, thus it is unknown whether the improved glucose metabolism is due to reduced body weight or an inherent insulin sensitivity change.

5) Citations 44 and 47 appear to be the same.

Reviewer #2 (Remarks to the Author):

In the current study, the authors investigated the effects of the mitochondria-derived peptide, SHLP2, in regulating the metabolism via its action on specific hypothalamic neurons. The authors provided evidence indicating that treatment with SHLP2 reduces the body weight and adiposity of mice by reducing food intake and increasing energy expenditure/thermogenesis. This effect is observed either after central or peripheral SHLP2 infusion. SHLP2 activates arcuate nucleus POMC neurons which could explain the metabolic effects of SHLP2. Inhibition of POMC neurons prevents SHLP2 effects. Finally, the authors identified the CXCR7 as a likely receptor that mediates SHLP2 effects. These findings are original,

relevant and provide the first evidence that mitochondria-derived peptide regulates hypothalamic neurocircuits to effect energy homeostasis. However, several aspects can be improved in the manuscript:

- 1) The sex of the volunteers or animals used in the experiments must be described in all experiments.
- 2) The differences in the SHLP2 immunoreactivity in the plasma between obese, diabetic and healthy volunteers is subtle and was overestimated in the manuscript. For example, there is no difference between obese and healthy patients. It is likely that the sample size is too small, so the authors are encouraged to increase sample size in this experiment.
- 3) Why this experiment was not done in diabetic and obese mice too? The evaluation of circulating SHLP2 levels in mouse models of obesity and diabetes (HFD, ob/ob or db/db mice, etc.), compared to lean controls, can improve the manuscript since the remaining data were obtained in murine models.
- 4) The most critical limitation of the present study is the control group in the experiments in which mice received SHLP2 injections. There is practically no description of how SHLP2 was synthesized (the authors just described that SHLP2 was commercially purchased from AnyGen company). Depending on how this peptide was synthesized, there may be a greater chance of having contaminants that can cause significant effects on metabolism, regardless of the peptide per se. For example, hormones derived from recombination in bacteria may contain small fractions of LPS or other components that can reduce food intake and body weight. Thus, the control group that received saline injections is not sufficient to guarantee the effects of the peptide. Thus, the authors need to describe the synthesis of the peptide in detail and add a control group to ensure that potential contaminants do not explain the observed effect.
- 5) Fig. 2h. Noradrenaline did not increase significantly as indicated by the authors. Likewise, insulin sensitivity (Fig. 4j) seems to be very similar between the groups (please, analyze the AUC). The authors are overestimating these results.
- 6) Fig. 5. Activation of POMC neurons per se should not produce significant effects on metabolism, independently of SHLP2 peptide? Any potential effect of POMC neuronal activation was not described and I believe it would be expected.
- 7) The authors inhibited POMC neurons during SHLP2 peptide treatment to prove that this neuronal population mediates the effects of SHLP2. Although blocking POMC neurons appears to prevent SHLP2-induced effects, the absence of effects may simply be a result of blocking the action of POMC neurons that induce satiety. A complementary approach should be used to confirm these results. For example, if AgRP neurons were manipulated, is the effect of SHLP2 maintained?

Reviewer #3 (Remarks to the Author):

The manuscript entitled "A mitochondria-derived peptide, SHLP2, regulates energy homeostasis through the activation of hypothalamic POMC neurons" by Professor Kim and colleagues describes that SHLP2, a mitochondria-derived peptide, suppress food intake and promoted thermogenesis via POMC neuron in ARC of hypothalamus. In addition, authors identified CXCR7 as the specific receptor for SHLP2. The experiments are well conducted, and methods, results and discussion are well written, however there are several significant concerns that need to be addressed.

Comments,

1. Serum SHLP2 was measured by dot blotting, and data showed that serum SHLP2 levels increased in obese and diabetic patients. As described in "introduction", SHLP 1 to 6 were discovered, thus SHLP2 antibody specificity or cross-reactivity must be shown in this paper. SHLP2 antibody generated in this study recognize other SHLPs? In addition, why authors used dot blotting to measure the concentration? As shown in Extended data Fig. 1, western blotting should be more specific rather than dot blotting. Also, to estimate the concentration, standard curve should be shown because you have synthetic SHLP2 for IP and ICV injection study.
2. Antigen information of SHLP2 antibody should be shown.
3. Where is the major SHLP2 source of tissue or organ? There are difference between IP and ICV injection of SHLP2 on c-fos expression in the hypothalamus. There is no difference at LH in IP injection but significant increase of c-fos at LH in ICV injection. Which is close to physiological or pathophysiological condition? SHLP2 are produced in the brain?
4. Distribution of CXCR7 in the brain should be shown by in situ hybridization or immunohistochemistry or qPCR using Laser micro-dissected nuclei of the brain. Concerning the data that SHLP2 increase thermogenesis, CXCR7 are expressed in the adipose tissue? Or, POMC neuron promote thermogenesis? Mechanisms should be discussed. Also, if this has been already published, authors should discuss the CXCR7 tissue distribution and the effect of SHLP2 in this study.
5. Cross-reactivity by SHLP1-6 on CXCR7 activation are required to be shown.
6. EC50 of SHLP2 against CXCR7 is about 1uM, indicating that plasma or serum concentrations would be high. What is the plasma or concentration of SHLP2? Similar question to No.1, the information of plasma concentration in normal, obese, and diabetic patients and animal model would be important, thus RIA or ELISA or alternative methods is better than dot blot analysis.

Reviewer #4 (Remarks to the Author):

In the study, the authors found that SHLP2 protects from high-fat diet-induced obesity and improves insulin sensitivity. These effects are likely due to suppressed food intake and increased thermogenesis. Interestingly, electrophysiological and pharmacogenetic approaches revealed that SHLP2 activates POMC neurons, and the activation is necessary for the anti-obesity effect of the peptide. Furthermore, the authors identified CXCR7 as a potential receptor for SHLP2, and the CXCR7 signaling axis is required for POMC neuron activation. Thus, the authors establish a new role for SHLP2 in energy balance, which is mediated via CXCR7 signaling, although it has not been suspected such roles. The overall data are sound and novel. However, the current manuscript can be further improved by addressing the reviewer's comments.

Comments.

The authors used 2mg/kg of SHLP2 for IP injection and showed the anti-obesity effect. How did the authors determine the concentration of SHLP2 for the injection? Similarly, explain how the authors determine the SHLP2 concentration for ICV injection. Do they have any dose-response effects?

The DREADD experiment shown in Fig.5h seems essential to suggest the requirements of POMC neurons for SHLP2's effect on food intake suppression. To enhance scientific rigor, the authors need to show the validation of this approach. They can show all the mouse validation for AAV-hM4Di injection into the POMC-Cre mice.

The authors used a chemical blocker to validate the requirement of CXCR7 for SHLP2's effect. This approach is a piece of good evidence to suggest the potential necessity for CXCR7 on POMC neurons or food intake regulation by SHLP2. However, the authors cannot rule out the off-target effects of the blocker. It would be informative if the authors could use a specific CXCR7 KO approach to show the specific evidence showing that the SHLP2 – CXCR7 signaling axis is crucial in regulating energy homeostasis.

For their metabolic cage studies, they need to clarify their analytic methods for energy expenditure either in "material and method" or "figure legend."

Showing their original blots for Figs.2i, 4h, 6d, and 6g would be good to clarify their findings.

The lack of knowledge regarding the role of SHLP2 in normal physiology conditions is a limitation of the current study. They only studied the effects of exogenous Shlp2 peptide (IP and ICV injection) on energy metabolism. The authors can discuss this issue in the manuscript.

Theoretically, SHLP2 can be generated from organs, cells, and neurons. Thus, it is possible that brain-born SHLP2 could directly activate the POMC neuronal activation. At the same time, peripheral-born SHLP2 is also played a role in the hypothalamic neurons. How the authors could distinguish the central vs peripheral effects of the peptide on energy balance. These issues can be discussed in the manuscript.

Response Letter (NCOMMS-22-36364-T)

We sincerely appreciate the Editor and Reviewers for taking your valuable time to carefully read and consider our work. Here, we would like to address the Reviewer's comments as follows:

REVIEWER COMMENTS

Reviewer #1 (Remarks to the Author):

This study investigated the role of SHLP2, a mitochondrial gene derived peptide, in DIO and the potential underlying mechanism. The data presented showed that SHLP2 reduced DIO, improved glucose metabolism, and activated a subset of brain neurons, including POMC neurons. Further studies suggested that it may be mediated by CLCX7 receptors. Although SHLP2 has been previously suggested to produce beneficial effects on metabolism, this study further extended that by examining its effect on DIO and with brain-specific effects. The experiments appeared to be vigorously investigated with all necessary controls and data were also appropriately analyzed. However, the evidence with POMC neurons as the mediating mechanism seems not be strong.

1. The study proposed that SHLP2 inhibited feeding through activation of feeding. However, previous studies including the one Zhan C et al, J. Neurosci. 2013 suggest that optogenetic activation of POMC neurons fails to reduce feeding, especially within the a few hours time window. Other studies with Gq Dredd also showed the same result. Thus, it is perplexing that SHLP2 activation of POMC neurons was able to produce such an impressive effect on reducing feeding within a 4 hours period. The authors need to conduct additional experiments to clarify this critical issue.

Response: As the reviewer mentioned, various research groups have demonstrated different effects on food intake suppression resulting from the activation of POMC neurons. For example, Zhan et al. activated POMC neurons in either the ARC or NTS regions using the DREADD system and found that NTS POMC neuron activation instantly inhibited feeding, while hypothalamic POMC neurons were necessary for long-term suppression¹. Similarly, Jiang et al. observed a reduction in food intake 24 hours after POMC neuron activation². In contrast, Wei et al. observed food intake suppression within 30 minutes after POMC neuron activation³.

This study involved mice that had been fasted for 4 hours, while the other two studies did not fast the mice³. These findings suggest that the time window for food intake regulation by POMC neurons may be different in fed and fasted conditions. In our study, we fasted mice overnight and administered SHLP2 to activate ARC POMC neurons, which led to a reduction in acute food intake for 4 hours.

Overnight fasting can significantly activate the AgRP/NPY neurons in the ARC^{4,5}. This may result in different effects on feeding suppression when POMC neurons are activated. As the reviewer suggested in the next comment, feeding suppression by SHLP2 could be partly mediated by its potential inhibitory effect on the AgRP/NPY neurons.

To address this, we conducted whole-cell patch-clamp experiments to evaluate the effects of SHLP2 on AgRP neuron activity in AgRP-cre:Td-Tomato mice. We found that SHLP2 application marginally inhibited AgRP neuron activity, suggesting that the inhibitory effect of SHLP2 on AgRP/NPY neurons may partly contribute to its effects on feeding. Accordingly, we have updated the Results and Discussion sections to incorporate these findings and also included them in Supplementary Fig. 5c, d.

2. It is also important that the effects of SHLP2 may be mediated by other neurons that are activated by SHLP2, as the authors showed additional brain regions within the Arc as well as other brain sites with c-Fos activation by SHLP2. Importantly, some of the effect of SHLP2 may be mediated by neuron inhibition, which is suggested by, for example, reduced AgRP/NPY shown in this manuscript, but can't be detected with c-Fos used here. Thus, the effect on POMC neurons seems to be, at most, one component of the mechanism.

Response: Thank you for the insightful comment. In respect to the reviewer's comment, we conducted a whole-cell patch clamp experiment to investigate SHLP's effect on AgRP neurons. As discussed in Question #1, our results showed that SHLP2 had a marginal inhibitory effect on AgRP neuron activity. This suggests that the suppression of AgRP/NPY neurons by SHLP2 may contribute at least in part to its effect on feeding. Therefore, we have included the electrophysiology data in the Supplementary Fig. 5c, d. We appreciate the reviewer's valuable and constructive suggestion.

3. Although the biology of SHLP2 on CLCX7 was well characterized, how it is related to POMC neurons is completely lacking. Thus, the description on mechanism related to POMC

neurons is not well supported by the data.

Response: We acknowledged the reviewer for this insightful comment. To address reviewer's comments and to provide further understanding of the CXCR7-POMC relationship, we conducted additional analysis of CXCR7 expression in POMC neurons through double immunostaining in the brains of POMC-GFP mice. We found that CXCR7 is expressed in the majority of POMC neurons (Supplementary Fig. 6h). Next, we monitored the effect of ERK inhibitor on SHLP2-induced POMC excitability since the ERK is activated through SHLP2's agonism on CXCR7 (Fig. 6h, i). In this patch-clamp study, we found that ERK activation by SHLP2's agonism on CXCR7 is necessary for SHLP2-induced POMC excitability (Fig. 6h, i). This suggests that POMC activation by SHLP2 is required for CXCR7-mediated ERK activation. To further clarify this relationship, we are now generating POMC-specific CXCR7 KO mice and will examine the specific roles of CXCR7 function in POMC as a mediator of SHLP2's effects. Accordingly, we have updated the "Discussion" section to describe the potential future direction of using the POMC-specific CXCR7 KO mouse model.

4. The effect on glucose metabolism was shown in DIO mice with existing body weight difference, thus it is unknown whether the improved glucose metabolism is due to reduced body weight or an inherent insulin sensitivity change.

Response: This is an important question to determine whether the effects of SHLP2 on glucose metabolism are due to differences in body weight or its potential effect on insulin sensitivity. Therefore, we set up new cohort of mice that were fed either a normal chow diet (NC) or a high-fat diet (HFD) and treated with SHLP2 for three weeks. We then performed GTT and ITT tests to investigate the effects of SHLP2 on glucose metabolism. In NC-fed mice, SHLP2 treatment did not affect body weight, but significantly improved glucose tolerance and insulin sensitivity (Supplementary Fig. 2d-h). In HFD-fed mice, we performed GTT and ITT before body weight divergence and found that SHLP2 administration also exhibited improved GTT and ITT (Fig. 1j, k). These combined results suggest that the effects of SHLP2 on glucose metabolism in HFD-fed mice are not solely due to differences in body weight but may also result from its intrinsic effects on insulin sensitivity. Therefore, we have included these results and updated the Results and Discussion sections accordingly. We thank the reviewer for raising this important question.

5. Citations 44 and 47 appear to be the same.

Response: We apologize to the reviewer for this mistake and have corrected these citations in the revised manuscript.

Reviewer #2 (Remarks to the Author):

In the current study, the authors investigated the effects of the mitochondria-derived peptide, SHLP2, in regulating the metabolism via its action on specific hypothalamic neurons. The authors provided evidence indicating that treatment with SHLP2 reduces the body weight and adiposity of mice by reducing food intake and increasing energy expenditure/thermogenesis. This effect is observed either after central or peripheral SHLP2 infusion. SHLP2 activates arcuate nucleus POMC neurons which could explain the metabolic effects of SHLP2. Inhibition of POMC neurons prevents SHLP2 effects. Finally, the authors identified the CXCR7 as a likely receptor that mediates SHLP2 effects. These findings are original, relevant and provide the first evidence that mitochondria-derived peptide regulates hypothalamic neurocircuits to effect energy homeostasis. However, several aspects can be improved in the manuscript

1. The sex of the volunteers or animals used in the experiments must be described in all experiments.

Response: We apologize to the reviewer for lacking the gender information of experimental animals and human subjects. We have now included this information in the revised manuscript. We appreciate the reviewer for this comment.

2. The differences in the SHLP2 immunoreactivity in the plasma between obese, diabetic and healthy volunteers is subtle and was overestimated in the manuscript. For example, there is no difference between obese and healthy patients. It is likely that the sample size is too small, so the authors are encouraged to increase sample size in this experiment.

Response: Following the reviewer's suggestion, we increased the sample size to 6 to 7 people and used only male samples due to the unavailability of enough female healthy controls, as

described in Fig. 1a. In the revised experiment with increased sample size, we found that diabetic and obese patients had significantly lower serum SHLP2 levels compared to healthy individuals. We have incorporated these results into Fig. 1a and updated the Results and Discussion sections accordingly.

3. Why this experiment was not done in diabetic and obese mice too? The evaluation of circulating SHLP2 levels in mouse models of obesity and diabetes (HFD, *ob/ob* or *db/db* mice, etc.), compared to lean controls, can improve the manuscript since the remaining data were obtained in murine models.

Response: As the reviewer suggested, we conducted dot blot analysis to measure SHLP2 levels in *ob/ob* and *db/db* mice. Interestingly, the results were consistent with those obtained from human serum, showing a significant decrease in SHLP2 levels in both *ob/ob* and *db/db* mice. We have included these findings in Supplementary Fig. 1e and updated the manuscript accordingly.

4. The most critical limitation of the present study is the control group in the experiments in which mice received SHLP2 injections. There is practically no description of how SHLP2 was synthesized (the authors just described that SHLP2 was commercially purchased from AnyGen company). Depending on how this peptide was synthesized, there may be a greater chance of having contaminants that can cause significant effects on metabolism, regardless of the peptide per se. For example, hormones derived from recombination in bacteria may contain small fractions of LPS or other components that can reduce food intake and body weight. Thus, the control group that received saline injections is not sufficient to guarantee the effects of the peptide. Thus, the authors need to describe the synthesis of the peptide in detail and add a control group to ensure that potential contaminants do not explain the observed effect.

Response: We apologized to the reviewer for lacking detailed information on the synthesis of the SHLP2 peptide. We now updated the “Material and Methods” section and provided detailed information on the peptide synthesis procedure. Briefly, SHLP2 was synthesized by a chemical synthesis method with stepwise deprotection of amino acids using 20% piperidine in dimethylformamide (DMF) and coupling of amino acids using Hexafluorophosphate Benzotriazole Tetramethyl Uronium (HBTU), N-methylmorpholine (NMM) in DMF. Synthesized peptides were then purified by a fractionation method run by HPLC using commercial columns (YMC-Triart C18/S-5 $\mu\text{m}/12\text{nm}.5\mu\text{m}$ (20 x 250 mm)). The fractions

containing the pure peptide were mixed and then lyophilized.

Next, we examined the effect of synthesized SHLP2 on feeding and body weight regulation. As the reviewer suggested, we included scrambled peptide and vehicle (saline) controls in our experiments. Our findings showed that unlike the vehicle and scrambled peptide, SHLP2 significantly suppressed food intake and body weight gain, confirming the specific role of SHLP2. Therefore, we have added these data to Supplementary Fig. 2b, c and updated the "Results" and "Materials and Methods" sections in the revised manuscript. We appreciate the reviewer for the comment.

5. Fig. 2h. Noradrenaline did not increase significantly as indicated by the authors. Likewise, insulin sensitivity (Fig. 4j) seems to be very similar between the groups (please, analyze the AUC). The authors are overestimating these results.

Response: To confirm whether SHLP2 can alter NE levels, we prepared an additional cohort and increased the sample size from 5 to 10 for each genotype. The results showed a significant increase in plasma NE levels in the SHLP2-treated mice (Fig. 2h).

As suggested by the reviewer, we measured the area under the curve (AUC) of the insulin tolerance test (ITT) in Fig. 4j and found no significant difference between the two groups.

We have added the results in Fig. 2h and Fig. 4i, j, respectively, and updated the manuscript to clarify the effect of SHLP2 administered via ICV, as suggested by the reviewer. Furthermore, we have further discussed the difference between ICV and IP administration of SHLP2 on insulin sensitivity to better clarify our findings. We appreciate the reviewer's important comment.

6. Fig. 5. Activation of POMC neurons per se should not produce significant effects on metabolism, independently of SHLP2 peptide? Any potential effect of POMC neuronal activation was not described and I believe it would be expected.

Response: Several metabolic consequences are related to the activation of POMC neurons, such as food intake, thermogenesis, leptin sensitivity, and iBAT function. Our current results imply that SHLP2 might functionally be involved in regulating these metabolic consequences. In addition, intrinsic POMC neurons also regulate other metabolic consequences, such as blood pressure and cardiac function. Therefore, it would be interesting to examine whether SHLP2 is

also involved in the regulation of these specific physiological functions. Therefore, in the revised manuscript, we added a discussion of the intrinsic POMC neuron effect that is independent of SHLP2 to the 'Discussion' section.

7. The authors inhibited POMC neurons during SHLP2 peptide treatment to prove that this neuronal population mediates the effects of SHLP2. Although blocking POMC neurons appears to prevent SHLP2-induced effects, the absence of effects may simply be a result of blocking the action of POMC neurons that induce satiety. A complementary approach should be used to confirm these results. For example, if AgRP neurons were manipulated, is the effect of SHLP2 maintained?

Response: We appreciate the reviewer's comment. As the reviewer noted, it is possible that the activity of AgRP neurons might be involved in the observed effects of SHLP2. Therefore, we set up the new experiments and examined the effect of SHLP2 on AgRP/NPY neurons and found that it marginally inhibited their activity. This result suggests that SHLP2's effects on feeding behavior might be partly attributed to its inhibitory effect on AgRP/NPY neurons. Although we were unable to perform AgRP/NPY neuron activation experiments using DREDD or optogenetics due to time constraints, we believe that the AgRP/NPY neurons may still contribute to SHLP2's effect, at least to some extent. We have incorporated this new result in the Supplementary Fig. 5c, d and updated the Results and Discussion sections of the revised manuscript.

Reviewer #3 (Remarks to the Author):

The manuscript entitled "A mitochondria-derived peptide, SHLP2, regulates energy homeostasis through the activation of hypothalamic POMC neurons" by Professor Kim and colleagues describes that SHLP2, a mitochondria-derived peptide, suppress food intake and promoted thermogenesis via POMC neuron in ARC of hypothalamus. In addition, authors identified CXCR7 as the specific receptor for SHLP2. The experiments are well conducted, and methods, results and discussion are well written, however there are several significant concerns that need to be addressed.

Comments,

1. Serum SHLP2 was measured by dot blotting, and data showed that serum SHLP2 levels increased in obese and diabetic patients. As described in "introduction", SHLP 1 to 6 were discovered, thus SHLP2 antibody specificity or cross-reactivity must be shown in this paper. SHLP2 antibody generated in this study recognize other SHLPs? In addition, why authors used dot blotting to measure the concentration? As shown in Extended Data Fig. 1, western blotting should be more specific rather than dot blotting. Also, to estimate the concentration, standard curve should be shown because you have synthetic SHLP2 for IP and ICV injection study.

Response: We appreciate the reviewer for the critical questions. We summarized each response as below.

1-1) As described in "introduction", SHLP 1 to 6 were discovered, thus SHLP2 antibody specificity or cross-reactivity must be shown in this paper. SHLP2 antibody generated in this study recognize other SHLPs?

Response: As shown in Supplementary Fig. 1d, we performed Western blots and confirmed the specificity of SHLP2 antibody.

1-2) In addition, why authors used dot blotting to measure the concentration? As shown in Extended data Fig. 1, western blotting should be more specific rather than dot blotting.

Response: We agree with the reviewer that Western blotting is a more precise method to detect serum SHLP2 level. Therefore, we had tried several times with different polyacrylamide gels but could not detect endogenous SHLP2 using conventional

Western blot not only in the human serum but also in the cell culture system. We can only detect when we conjugate SHLP2 with GFP and overexpress the SHLP2-GFP construct to a cell culture system (Supplementary Fig. 1a) or when we directly load SHLP2 peptide lysates (Supplementary Fig. 1b-d).

It is very challenging to detect endogenous small peptides using conventional Western blotting in part by the small peptides easily detached from the blotting membrane. To solve the problem, we even adapted the vacuum-assisted detection method, but unfortunately endogenous SHLP2 was not detected⁶. Therefore, we used dot blotting. Since serum contains a large amount of non-specific proteins (e.g. albumin, globulins,...), we used the serial dilution method to minimize the interference of these non-specific proteins. We created a standard curve for each run, but we did not use it to measure the concentration because dot blot is typically used for relative concentration rather than absolute value, mainly due to its lower sensitivity.

2. Antigen information of SHLP2 antibody should be shown.

Response: We apologize for the insufficient description in the Material and Method. This is a similar question raised by reviewer 2 (Question 4), and we have incorporated the updated information in the revised manuscript.

3. Where is the major SHLP2 source of tissue or organ? There are difference between IP and ICV injection of SHLP2 on c-fos expression in the hypothalamus. There is no difference at LH in IP injection but significant increase of c-fos at LH in ICV injection. Which is close to physiological or pathophysiological condition? SHLP2 are produced in the brain?

Response: Although Cobb *et al.* reported SHLP2 expressed in liver, kidney, muscles and brain, further studies determining the exact tissue or organ producing SHLP2 especially in human would be required⁷. In addition, the distinct levels of c-Fos activation in the hypothalamic regions (Fig. 3c-f) would be come from the route difference of SHLP2 administration. The IP injection might less stable and, thus, show less LH c-Fos activation than direct ICV administration.

4. Distribution of CXCR7 in the brain should be shown by in situ hybridization or immunohistochemistry or qPCR using Laser micro-dissected nuclei of the brain. Concerning the data that SHLP2 increase thermogenesis, CXCR7 are expressed in the adipose tissue? Or,

POMC neuron promote thermogenesis? Mechanisms should be discussed. Also, if this has been already published, authors should discuss the CXCR7 tissue distribution and the effect of SHLP2 in this study.

Response: We thank the reviewer for the insightful comment. As the reviewer indicated, expression pattern of CXCR7 in mouse brain was established⁸. In the mouse brain, the CXCR7 is expressed in hypothalamus, cerebral cortex, hippocampus, subventricular zone, ventricular walls, cerebellum and spinal cord. It is also highly expressed in adipose tissue, liver and skeletal muscle.

It is well-known that activation of POMC neurons affects whole-body energy homeostasis not only by regulating feeding but also by affecting thermogenesis possible through the autonomic nervous system. We found that SHLP2 activates POMC neurons and it inhibited feeding and increased BAT thermogenesis. Therefore, we have updated the manuscript and incorporated the reviewer comments in revised manuscript.

5. Cross-reactivity by SHLP1-6 on CXCR7 activation are required to be shown.

Response: To address the reviewer's question, we have newly performed the Nanoluc-complementation assay to investigate whether other SHLPs have an effect on CXCR7 activation (Supplementary Fig. 6a). We found that CXCR7 was specifically activated by SHLP2, but not by SHLP1, 3, 4, 5, or 6. These results suggest that SHLP2 is a specific agonist for CXCR7. We appreciate to the reviewer's insightful comment, which helped us to clarify the specificity of SHLP2 on CXCR7.

6. EC50 of SHLP2 against CXCR7 is about 1uM, indicating that plasma or serum concentrations would be high. What is the plasma or concentration of SHLP2? Similar question to No.1, the information of plasma concentration in normal, obese, and diabetic patients and animal model would be important, thus RIA or ELISA or alternative methods is better than dot blot analysis.

Response: We agree with the reviewer that RIA or ELISA would be more precise methods for detecting serum SHLP2 levels. Regrettably, commercial RIA or ELISA kits for this purpose are not currently available. As a result, we were unable to provide an exact SHLP2 concentration in plasma due to the lack of commercial RIA or ELISA kits.

Reviewer #4 (Remarks to the Author):

In the study, the authors found that SHLP2 protects from high-fat diet-induced obesity and improves insulin sensitivity. These effects are likely due to suppressed food intake and increased thermogenesis. Interestingly, electrophysiological and pharmacogenetic approaches revealed that SHLP2 activates POMC neurons, and the activation is necessary for the anti-obesity effect of the peptide. Furthermore, the authors identified CXCR7 as a potential receptor for SHLP2, and the CXCR7 signaling axis is required for POMC neuron activation. Thus, the authors establish a new role for SHLP2 in energy balance, which is mediated via CXCR7 signaling, although it has not been suspected such roles. The overall data are sound and novel. However, the current manuscript can be further improved by addressing the reviewer's comments.

Comments.

1. The authors used 2mg/kg of SHLP2 for IP injection and showed the anti-obesity effect. How did the authors determine the concentration of SHLP2 for the injection? Similarly, explain how the authors determine the SHLP2 concentration for ICV injection. Do they have any dose-response effects?

Response: We determined the appropriate dose of SHLP2 for IP injection by conducting preliminary screening experiments on normal chow diet (NC)-feeding mice. We tested two SHLP2 doses, 2 mg/kg and 3 mg/kg, and found that both doses affected cumulative food intake. After careful consideration, we chose the 2 mg/kg dose for further experiments.

For the optimal dose for ICV administration, we examined the dose-dependent effects of SHLP2 on cumulative food intake and body weight. We tested several ICV doses of SHLP2, including 0.5 μ g, 1 μ g, and 3 μ g, and found that the administration of 3 μ g of ICV SHLP2 produced significant effects on food intake and body weight. Therefore, we selected this dose for further experiments. To ensure the appropriate selection of SHLP2 dosing in our study, we have updated the manuscript to include the results of our experiments in Supplementary Fig. 2a and Supplementary Fig. 4b, c. We are thankful to the reviewer for bringing this to our attention.

2. The DREADD experiment shown in Fig.5h seems essential to suggest the requirements of POMC neurons for SHLP2's effect on food intake suppression. To enhance scientific rigor, the authors need to show the validation of this approach. They can show all the mouse validation

for AAV-hM4Di injection into the POMC-Cre mice.

Response: We thank the reviewer for this suggestion. We have provided all of data images validating the AAV-virus injection into the POMC-cre mice in the Source Data Fig. 5i.

3. The authors used a chemical blocker to validate the requirement of CXCR7 for SHLP2 2's effect. This approach is a piece of good evidence to suggest the potential necessity for CXCR7 on POMC neurons or food intake regulation by SHLP2. However, the authors cannot rule out the off-target effects of the blocker. It would be informative if the authors could use a specific CXCR7 KO approach to show the specific evidence showing that the SHLP2 – CXCR7 signaling axis is crucial in regulating energy homeostasis.

Response: We agree with the reviewer's opinion that a genetic knock-out approach will strengthen the specific role of CXCR7 in mediating SHLP2 actions. To this end, we are currently generating CXCR7 KO mice specifically in the POMC neurons using floxed CXCR7 and POMC-Cre mouse lines. However, generating KO mice and analyzing the model will take some time. Therefore, to address the reviewer's suggestion, we have updated the "Discussion" section and introduced tissue-specific CXCR7 KO as a future direction.

4. For their metabolic cage studies, they need to clarify their analytic methods for energy expenditure either in “material and method” or “figure legend.”

Response: We have clarified the analytical methods for measuring the energy expenditure in the Materials and Methods section of the revised manuscript. In brief, we normalized energy expenditure using either lean body mass or body weight^{0.75}. We used the latter because lean mass could not be measured due to the presence of the ICV cannula. We thank to the reviewer for this constructive suggestion.

5. Showing their original blots for Figs.2i, 4h, 6d, and 6g would be good to clarify their findings.

Response: We have included all the original uncropped blots in the “Source Data”.

6. The lack of knowledge regarding the role of SHLP2 in normal physiology conditions is a limitation of the current study. They only studied the effects of exogenous Shlp2 peptide (IP and ICV injection) on energy metabolism. The authors can discuss this issue in the manuscript.

Response: We acknowledge to the reviewer for the insightful comment. As the reviewer

suggested, we have updated the Discussion part in the revised manuscript and added current knowledge on the normal physiology of SHLP2.

7. Theoretically, SHLP2 can be generated from organs, cells, and neurons. Thus, it is possible that brain-born SHLP2 could directly activate the POMC neuronal activation. At the same time, peripheral-born SHLP2 is also played a role in the hypothalamic neurons. How the authors could distinguish the central vs peripheral effects of the peptide on energy balance. These issues can be discussed in the manuscript.

Response: We agree with the reviewer's view that it would be important to know region-specific autocrine or paracrine effect of the SHLP2 together with endocrine effects. We consider the tissue-specific KO approaches might, at least in part, be helpful to understand central or peripheral effect of SHLP2. To incorporate the reviewer's opinion, we revised the discussion section and further discuss this possibility. We appreciate the reviewer for this insightful comment.

References

- 1 Zhan, C. *et al.* Acute and long-term suppression of feeding behavior by POMC neurons in the brainstem and hypothalamus, respectively. *The Journal of neuroscience : the official journal of the Society for Neuroscience* **33**, 3624-3632, doi:10.1523/JNEUROSCI.2742-12.2013 (2013).
- 2 Jiang, J., Morgan, D. A., Cui, H. & Rahmouni, K. Activation of hypothalamic AgRP and POMC neurons evokes disparate sympathetic and cardiovascular responses. *American journal of physiology. Heart and circulatory physiology* **319**, H1069-H1077, doi:10.1152/ajpheart.00411.2020 (2020).
- 3 Wei, Q. *et al.* Uneven balance of power between hypothalamic peptidergic neurons in the control of feeding. *P Natl Acad Sci USA* **115**, E9489-E9498, doi:10.1073/pnas.1802237115 (2018).
- 4 Aponte, Y., Atasoy, D. & Sternson, S. M. AGRP neurons are sufficient to orchestrate feeding behavior rapidly and without training. *Nature neuroscience* **14**, 351-355, doi:10.1038/nn.2739 (2011).
- 5 Wu, Q. *et al.* The temporal pattern of cfos activation in hypothalamic, cortical, and brainstem nuclei in response to fasting and refeeding in male mice. *Endocrinology* **155**, 840-853, doi:10.1210/en.2013-1831 (2014).
- 6 Tomisawa, S. *et al.* A new approach to detect small peptides clearly and sensitively by Western blotting using a vacuum-assisted detection method. *Biophysics (Nagoya-shi)* **9**, 79-83, doi:10.2142/biophysics.9.79 (2013).
- 7 Cobb, L. J. *et al.* Naturally occurring mitochondrial-derived peptides are age-dependent regulators of apoptosis, insulin sensitivity, and inflammatory markers. *Aging (Albany NY)* **8**, 796-809, doi:10.18632/aging.100943 (2016).
- 8 Banisadr, G., Podojil, J. R., Miller, S. D. & Miller, R. J. Pattern of CXCR7 Gene Expression in Mouse Brain Under Normal and Inflammatory Conditions. *J Neuroimmune Pharmacol* **11**, 26-35, doi:10.1007/s11481-015-9616-y (2016).

REVIEWERS' COMMENTS

Reviewer #1 (Remarks to the Author):

The authors have largely addressed the earlier concerns except for the evidence supporting POMC neurons in the action. The overall supporting evidence is largely based on pharmacology, which doesn't speak to the necessity of POMC neuron action. The only data supporting a role of POMC neurons is the hM4Di experiment, which itself is a bit problematic as the control experiment with hM4Di showed no differences in feeding or body weight upon POMC neuron inhibition. This is in contrast to the prevalent literature showing loss of function in POMC neurons causing hyperphagia and body weight increase. As suggested previously (PMID: 23426689), POMC neuron activation causes little effects on feeding/body weight, which is also supported by a recent publication (PMID: 37069175). Therefore, the authors are strongly suggested to tone down the contribution from POMC neurons or provide additional more compelling evidence demonstrating an essential function of POMC neurons.

Reviewer #2 (Remarks to the Author):

The authors did a great job satisfactorily answering to all my comments and performing the additional control experiments requested in my initial revision. I carefully revised the new version of the manuscript and I do not have any further suggestion. I believe the manuscript is now suitable for publication and it reveals important and novel findings regarding the metabolic effects of SHLP2 and its potential application for the treatment of obesity.

Reviewer #3 (Remarks to the Author):

The authors have answered all questions correctly and I consider this paper acceptable.

Reviewer #4 (Remarks to the Author):

The authors addressed all of the previous comments.

Response Letter (NCOMMS-22-36364A)

We would like to thank the Editor and Reviewers for their careful reading and thoughtful comments on our work. In response to the Reviewer's comments, we would like to address them as follows.

REVIEWER COMMENTS

Reviewer #1 (Remarks to the Author):

The authors have largely addressed the earlier concerns except for the evidence supporting POMC neurons in the action. The overall supporting evidence is largely based on pharmacology, which doesn't speak to the necessity of POMC neuron action. The only data supporting a role of POMC neurons is the hM4Di experiment, which itself is a bit problematic as the control experiment with hM4Di showed no differences in feeding or body weight upon POMC neuron inhibition. This is in contrast to the prevalent literature showing loss of function in POMC neurons causing hyperphagia and body weight increase. As suggested previously (PMID: 23426689), POMC neuron activation causes little effects on feeding/body weight, which is also supported by a recent publication (PMID: 37069175). Therefore, the authors are strongly suggested to tone down the contribution from POMC neurons or provide additional more compelling evidence demonstrating an essential function of POMC neurons.

Response: We agree with the reviewer's suggestion to reduce the emphasis on "POMC neurons" throughout the manuscript. Consequently, we have carefully made the necessary changes to present a broader perspective that acknowledges the potential involvement of other hypothalamic neurons. We sincerely appreciate the valuable feedback provided by the reviewer.

Reviewer #2 (Remarks to the Author):

The authors did a great job satisfactorily answering to all my comments and performing the additional control experiments requested in my initial revision. I carefully revised the new version of the manuscript and I do not have any further suggestion. I believe the manuscript is now suitable for publication and it reveals important and novel findings regarding the metabolic effects of SHLP2 and its potential application for the treatment of obesity.

Response: We appreciate the reviewer for providing valuable comments and feedback on our manuscript.

Reviewer #3 (Remarks to the Author):

The authors have answered all questions correctly and I consider this paper acceptable.

Response: We appreciate the reviewer for providing valuable comments and feedback on our manuscript.

Reviewer #4 (Remarks to the Author):

The authors addressed all of the previous comments.

Response: We appreciate the reviewer for providing valuable comments and feedback on our manuscript.